# A mathematical-descriptor of tumor-mesoscopic-structure from computed-tomography images annotates prognostic- and molecular-phenotypes of epithelial ovarian cancer

Haonan Lu [1,2], Mubarik Arshad[2], Andrew Thornton[1], Giacomo Avesani [2], Paula Cunnea [1], Ed Curry[1], Fahdi Kanavati[2], Jack Liang[2], Katherine Nixon[1], Sophie T. Williams[1], Mona Ali Hassan[1], David D.L. Bowtell[3,4], Hani Gabra[1,5], Christina Fotopoulou[1], Andrea Rockall[2,6,7] & Eric O. Aboagye [2]

The five-year survival rate of epithelial ovarian cancer (EOC) is approximately 35–40% despite maximal treatment efforts, highlighting a need for stratification biomarkers for personalized treatment. Here we extract 657 quantitative mathematical descriptors from the preoperative CT images of 364 EOC patients at their initial presentation. Using machine learning, we derive a non-invasive summary-statistic of the primary ovarian tumor based on 4 descriptors, which we name "Radiomic Prognostic Vector" (RPV). RPV reliably identifies the 5% of patients with median overall survival less than 2 years, significantly improves established prognostic methods, and is validated in two independent, multi-center cohorts. Furthermore, genetic, transcriptomic and proteomic analysis from two independent datasets elucidate that stromal phenotype and DNA damage response pathways are activated in RPV-stratified tumors. RPV and its associated analysis platform could be exploited to guide personalized therapy of EOC and is potentially transferrable to other cancer types.

[1] Ovarian Cancer Action Research Centre, Department of Surgery and Cancer, Faculty of Medicine, Imperial College London, London W12 0HS, UK. [2] Cancer Imaging Centre, Department of Surgery and Cancer, Faculty of Medicine, Imperial College London, London W12 0HS, UK. [3] Peter MacCallum Cancer Centre, Melbourne 3010 VIC, Australia. [4] Sir Peter MacCallum Department of Oncology, The University of Melbourne, Melbourne 3010 VIC, Australia. [5] Early Clinical Development, iMED Biotech Unit, AstraZeneca, Cambridge SG8 6HB, UK. [6] Department of Radiology, Imperial College Healthcare NHS Trust, London W12 0HS, UK. [7] Department of Radiology, The Royal Marsden NHS Foundation Trust, London SW3 6JJ, UK. These authors contributed equally: Haonan Lu, Mubarik Arshad. Correspondence and requests for materials should be addressed to E.O.A. (email: eric.aboagye@imperial.ac.uk)

"R" adiomics" quantifies mesoscopic tumor phenotype from anatomic or functional images by defining tumor spatial complexity—including first and higher order statistics, fractal and shape features—generating disease features not appreciated by the naked eye[1–3]. The development of a radiomics approach for disease phenotyping, using routine pre-surgical computed tomography (CT), as an extension of current imaging semantics is therefore promising[4–6].

Epithelial ovarian cancer (EOC) is the sixth most common cancer among women in the UK and has the highest mortality of all gynecological cancers, accounting for 4% of all cancer deaths in women[7]. High-grade serous ovarian cancer (HGSOC) represents the most dominant (70% of EOC patients) and most lethal histological subtype[8]. Although it is well known that HGSOC patients have a heterogeneous response to treatment and prognosis, extensive cytoreductive surgery combined with platinum-based chemotherapy are currently the standard treatments for most patients without consideration of individual prognostic and predictive biomarkers. Recently, a number of studies including the Cancer Genome Atlas (TCGA) project have obtained a comprehensive genomic profile of HGSOC, resulting in several molecular prognostic biomarker discoveries[9]. For instance, CCNE1 amplification is commonly associated with platinum-resistant and refractory disease[10,11]; HGSOCs were classified into prognostically distinct molecular subtypes according to gene expression profiling[12–14]. More recently, large sets of microRNAs have been exploited to determine the risk profile of EOC[15]. It remains challenging, however, to translate these molecularly determined characteristics into clinically relevant biomarkers due to intratumor heterogeneity, additional high assay cost, and time delays. Therefore, a noninvasive, real-time, and cost-effective prognostic marker approach is warranted to reliably guide personalized treatment of EOC patients.

In the current study, a novel radiomics-determined mathematical descriptor of EOC tumor risk phenotype with a reliable, convincing predictive value is discovered and validated, and further insights into the biological basis of the descriptor is provided through investigation of correlated transcriptomics, proteomics and copy-number alterations (CNAs).

## Results

**Characteristics of data and patients**. We developed TexLab 2.0, a software program that summarized 657 features relating to the shape and size, intensity, texture and wavelet decompositions of 364 preoperative contrast-enhanced CT scans[16] (Table 1 and Supplementary Figure 1). All the radiomic features are summarized in Supplementary Data 1. A comprehensive molecular profile including gene expression, copy-number, and protein expression was analyzed for a subset of patients (Table 1). The study workflow is summarized in Supplementary Figure 2.

**Table 1 Summary of data produced**

| Data type | Cohort | Platforms | Features | Cases |
|---|---|---|---|---|
| Radiomic profile | HH | TexLab 2.0 | 657 | 294 |
| | TCGA | | | 70 |
| DNA copy number | HH | Affymetrix SNP6 | Whole genome | 84 |
| | TCGA | | | 70 |
| Protein expression | HH | RPPA | 299 | 198 |
| | TCGA | | 199 | 48 |
| mRNA expression | HH | Illumina MiSeq | 68 | 173 |
| | TCGA | Affymetrix U133 | Whole genome | 70 |

*HH* Hammersmith Hospital, *TCGA* The Cancer Genome Atlas

We evaluated 294 primary EOC patients with fresh frozen tissue treated within the Hammersmith Hospital, Imperial College Healthcare NHS Trust, London, UK between 2004 and 2015 as well as 70 EOC patients from the TCGA project (Supplementary Table 1, Supplementary Figure 2).

**Overview of radiomic profile in epithelial ovarian cancer**. We wished to investigate the data structure within the radiomic profiles derived from primary tumors of EOC patients in relation to clinical and genetic features. For samples with both radiomics and CNA data, we performed a spectral clustering analysis based on the Pearson correlation coefficients between each samples' radiomic profile (Fig. 1a). There was a clear division of samples into three major groups with each group characterized by high feature similarity but largely distinct from those in other groups. Notably, one of these groups (Group 1) was found to be significantly enriched for HGSOC (Fig. 1b). EOC, particularly the HGSOC subtype, frequently features CNAs[17]. We found that Group 1 was enriched for tumors with high CNAs (Fig. 1c). This group had a worse outcome as measured by progression-free survival (PFS) (Supplementary Figure 3).

To further understand the radiomic characteristics of the HGSOC subtype, we performed unsupervised hierarchical clustering analysis using the radiomic profiles in the HH cohort. We found two distinct clusters within this population based purely on the radiomic profile (Fig. 1d). Cluster 2 was significantly associated with the presence of ascites ($p = 0.00729$, chi-squared test) and poor PFS ($p = 0.022$, log-rank test; Fig. 1e), marginally associated with higher tumor stage ($p = 0.0686$, Fisher's exact test), but not associated with postoperative residual disease or molecular subtype (Fig. 1d). Of interest, 96% of bilateral tumors from patient were assigned to the same cluster, revealing a close radiomic similarity (Supplementary Note 1).

In aggregate, unsupervised analysis highlighted an intrinsic association between radiomic profile, genetic background, and clinical characteristics, warranting further characterization.

**Radiomic prognostic vector predicts survival**. We used three datasets to assess the prognostic potential of the radiomic profile for HGSOC patients: HGSOC cases from the HH cohort were split into the HH discovery ($n = 136$) and the HH validation datasets ($n = 77$), and examined in parallel with the TCGA validation dataset ($n = 70$) (Supplementary Figure 2 and Supplementary Table 1). We firstly performed Cox regression with overall survival (OS) examining each radiomic feature in turn, using data from primary tumors in the HH discovery dataset (Supplementary Figure 2). Forty-two radiomic features were found to be significantly associated with OS (false discovery rate < 0.05; Fig. 2a; Supplementary Data 2). The 42 radiomic features were further reduced to 4 weighted features using least absolute shrinkage and selection operator (LASSO[18]) (Fig. 2b, c and Supplementary Table 2). The weighted sum of these four radiomic features gave a RPV score for each tumor.

With an unsupervised $k$-means clustering approach, we split all the patients from the three cohorts based on their RPV into three subgroups (low risk, medium risk, and high risk; Supplementary Table 3). The patient groups stratified by RPV had distinct OS differences in the discovery dataset ($N = 136$, $p < 0.0001$, log-rank test; Fig. 2d). Using the same RPV decision boundaries, OS differences were confirmed in two independent validation datasets, the TCGA validation dataset ($N = 70$, $p = 0.000105$, log-rank test; Fig. 2e) and the HH validation dataset ($N = 77$, $p = 0.0274$, log-rank test; Fig. 2f).

In a multivariable Cox regression model with age, stage, postoperative residual disease, neo-adjuvant chemotherapy, and

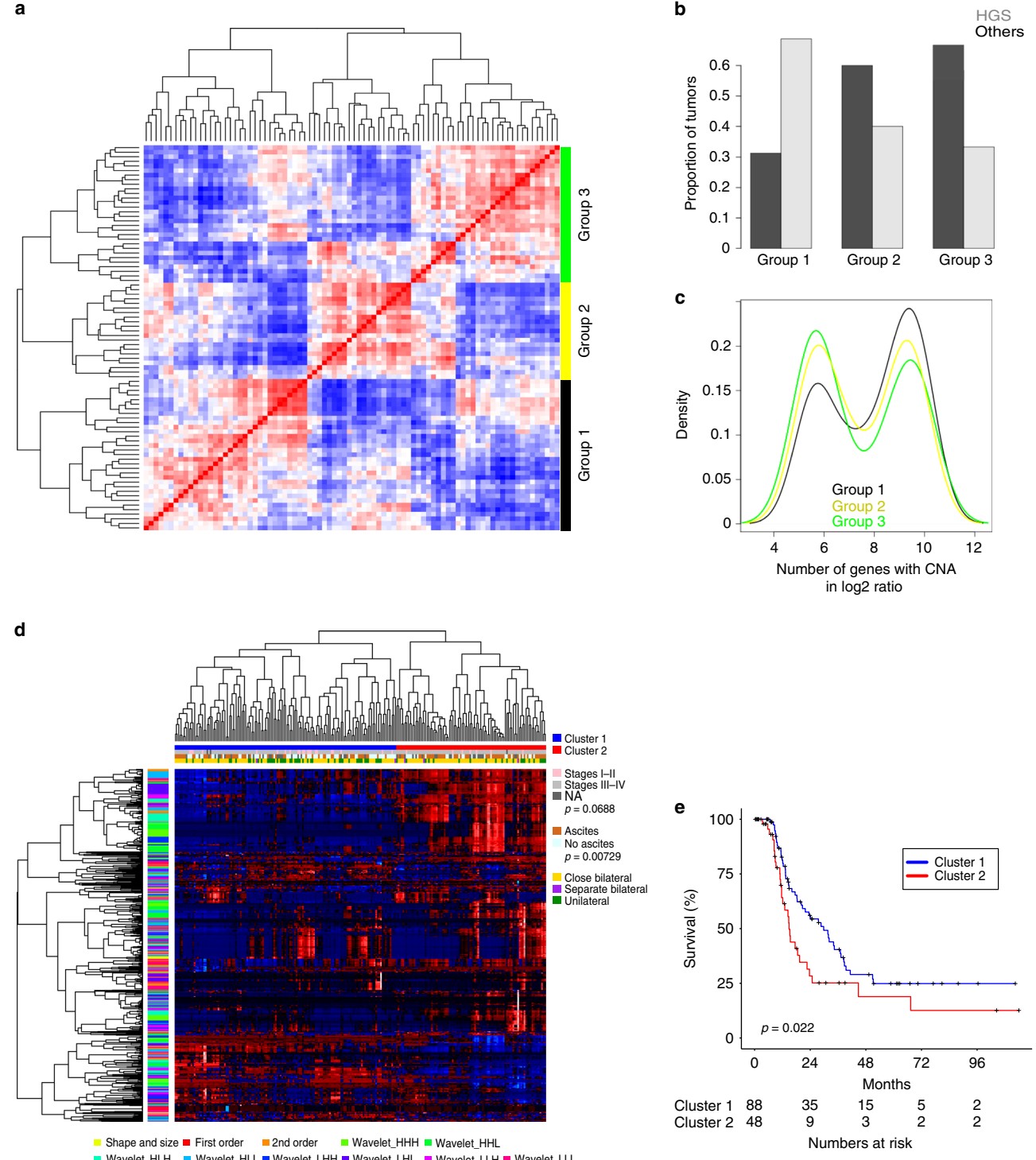

**Fig. 1** Unsupervised clustering analysis of radiomic data in EOC. **a** Heatmap illustrating clustered matrix of sample-wise similarities (HH cohort) based on whole tumor radiomic profiles of primary ovarian tumors from all EOC histology. Black bar, group 1; yellow bar, group 2; green bar, group 3. **b** Distribution of high-grade serous ovarian carcinomas over patient groupings defined by similarities of radiomic profile ($n = 84$, $p = 0.02$, Fisher's exact test). **c** Differences in the numbers of genes affected by copy-number aberration in tumors with spectral radiomic clusters. Black line, group 1; yellow line, group 2; green line, group 3. **d** Unsupervised hierarchical clustering of radiomic profile from primary HGSOC identified two distinct subgroups (blue and red as shown on the top row above heatmap). The associations between radiomic subgroups with the presence of ascites, lateral and tumor stage are indicated on the right. A summary of radiomic features are given on the y-axis. Blue bar, cluster 1; red bar, cluster 2. **e** Kaplan−Meier analysis of the radiomic subgroups with progression-free survival ($n = 136$). Blue line, cluster 1; red line, cluster 2. p value from log-rank test is included. HH Hammersmith Hospital, EOC epithelial ovarian cancer, HGSOC high-grade serous ovarian cancer

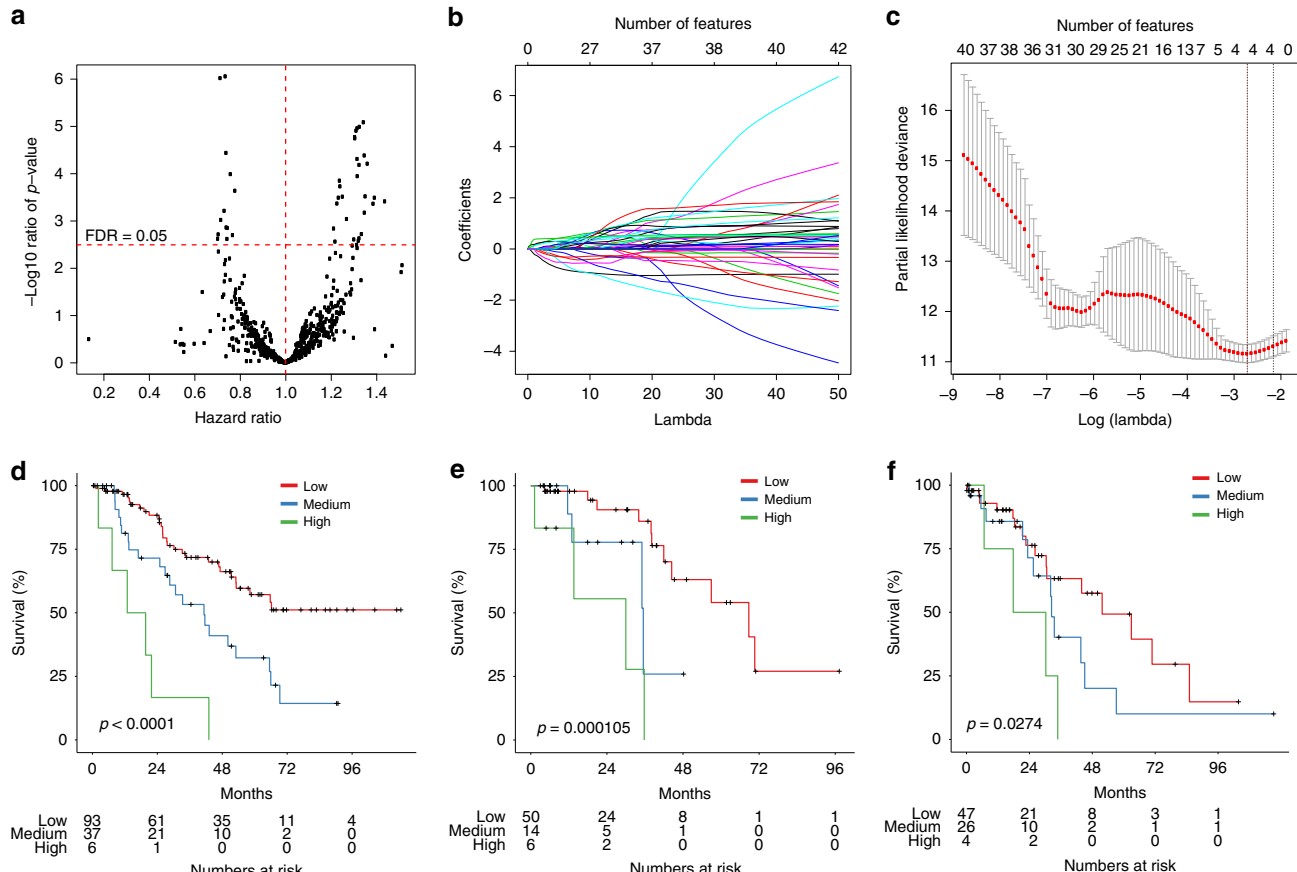

**Fig. 2** Prognostic model based on radiomic profile in HGSOC. **a** Summary of univariate Cox regression between each radiomic feature and overall survival in the discovery set. Each black point represents the *p* value (*y*-axis) and hazard ratio (HR; *x*-axis) of a radiomic feature. Red horizontal dashed line indicates the false discovery rate (FDR) of 0.05; red vertical dashed line indicates HR of 1. **b** Least absolute shrinkage and selection operator (LASSO) regression analysis was performed to select radiomic features for prognostic model-building for HGSOC patients. Feature coefficients were plotted against shrinkage parameter (Lambda). **c** Partial likelihood deviance from Cox regression (*y*-axis) was generated under different shrinkage parameters (*x*-axis). Number of features selected corresponding to each lambda are given above the plot. Kaplan−Meier analyses were performed between radiomic prognostic vector (RPV) and overall survival in **d** HH discovery cohort (*n* = 136), **e** TCGA validation cohort (*n* = 70) and **f** HH validation cohort (*n* = 77). Red line, RPV low; green line, RPV medium; blue line, RPV high. *p* values are given by log-rank test. HGSOC high-grade serous ovarian cancer, TCGA the Cancer Genome Atlas

the potential structured noise in the datasets (scan thickness), RPV remained significantly and continuously associated with OS in the discovery dataset (hazard ratio (HR): 3.83, 95% confidence interval (CI) (2.27–6.46), $p = 5.11 \times 10^{-7}$; RPV range: −0.322 to 3.16), as well as the TCGA validation dataset (HR: 4.87, 95% CI (1.67–14.2), $p = 0.00380$) and the HH validation dataset (HR: 7.36, 95% CI (1.29–41.9), $p = 0.0245$; Table 2). The addition of RPV improved the clinically available prognostic methods (stage, age, and postoperative residual disease) in all three datasets as measured by the concordance index (C-index)[19] (HH discovery: from 0.658 to 0.739; TCGA validation: from 0.549 to 0.690; HH validation: from 0.659 to 0.679). Age, stage, and postoperative residual disease were significantly associated with OS in either uni- or multivariable analysis in the combined HH cohort while RPV remained the strongest prognostic factor, suggesting RPV is prognostic in a representative HGSOC cohort. RPV was also found associated with OS independent of performance status in a subset of patients (Supplementary Table 4). We excluded performance status from the multivariable analysis to avoid misinterpretation in the presence of insufficient data, given that we only had the performance status of 62 out of the total 213 patients in the HH cohort, and less than 20 of them had a performance status >1. For that reason, any statistical

conclusions relating to performance status will not be valid due to the very small sample size. Notably, RPV possessed a better prognostic power when compared to the existing prognostic markers including CA125 and the transcriptome-based molecular subtype and potentially synergizes with existing CT-based morphological approaches (Supplementary Tables 5–7; Supplementary Note 1). Apart from prognosis, high RPV was found significantly associated with primary chemotherapy resistance, shorter PFS, and poor surgical outcome (Fig. 3e and Supplementary Figure 7g, Supplementary Note 1), suggesting RPV as a potential predictive marker in HGSOC.

Taking advantage of the gene expression profiles collected in parallel with radiomic profiles, we constructed a surrogate marker of RPV based on a weighted list of mRNA expressions in the TCGA validation dataset where both CT scans and gene expression profiles were available (eRPV; Supplementary Note 1). eRPV strongly correlated with RPV ($r = 0.720$) in the TCGA validation dataset and significantly interacted with RPV in the Cox regression model (Supplementary Figure 10c). It showed a similar prognostic potential as RPV in two additional cohorts (TCGA dataset without publicly available CT scans: $n = 448$, $HR = 2.19$, 95% CI (1.23–4.25), $p = 0.0208$; Tothill dataset: $n = 228$, $HR = 7.94$, 95% CI (2.02–31.3), $p = 0.00303$; adjusted

**Table 2 Summary of Cox regression analysis of RPV in three datasets. RPV was used as a continuous variable in the Cox regression analysis**

| | Variables | Univariate | | Multivariable | |
|---|---|---|---|---|---|
| | HR (95% CI) | p value | HR (95% CI) | p value | |
| HH discovery (n = 136) | RPV | 4.08 (2.48–6.71) | 3.37e-08 | 3.86 (2.30–6.46) | $3.04 \times 10^{-7}$ |
| | Stage | 2.03 (1.37–3.00) | 0.000426 | 1.88 (1.24–2.86) | 0.00305 |
| | Residual disease | 1.75 (1.03–2.99) | 0.0393 | 1.40 (0.803–2.44) | 0.235 |
| | Age[a] | 1.25 (0.741–2.11) | 0.404 | 1.47 (0.865–2.51) | 0.154 |
| HH validation (n = 77) | RPV | 2.05 (1.01–4.18) | 0.0485 | 5.08 (1.03–25.2) | 0.0465 |
| | Stage | 1.32 (0.775–2.24) | 0.309 | 1.32 (0.664–2.64) | 0.425 |
| | Residual disease | 1.78 (0.777–4.08) | 0.173 | 1.28 (0.514–3.21) | 0.593 |
| | Age[a] | 2.10 (0.940–4.68) | 0.0704 | 3.44 (1.19–9.94) | 0.0228 |
| HH cohort combined[b] (n = 213) | RPV | 2.94 (2.02–4.26) | $1.54 \times 10^{-8}$ | 3.32 (2.16–5.10) | $4.91 \times 10^{-8}$ |
| | Stage | 1.82 (1.33–2.48) | 0.00017 | 1.75 (1.24–2.50) | 0.0017 |
| | Residual disease | 1.72 (1.11–2.69) | 0.0163 | 1.36 (0.855–2.15) | 0.196 |
| | Age[a] | 1.46 (0.951–2.24) | 0.0835 | 1.74 (1.10–2.76) | 0.0183 |
| TCGA validation (n = 70) | RPV | 4.94 (2.06–11.8) | 0.00034 | 6.21 (2.06–18.7) | 0.00117 |
| | Stage | 1.75 (0.913–3.34) | 0.0921 | 1.03 (0.309–3.44) | 0.960 |
| | Residual disease | 1.34 (0.480–3.74) | 0.576 | 1.45 (0.414–5.05) | 0.564 |
| | Age[a] | 1.08 (0.435–2.66) | 0.874 | 0.500 (0.154–1.63) | 0.249 |

HR hazard ratio, CI confidence interval, RPV radiomic prognostic vector, HH Hammersmith Hospital, TCGA the Cancer Genome Atlas
[a]Age has been dichotomized at 60 years
[b]Combining HH discovery and HH validation datasets

for stage, grade, residual disease, age and neo-adjuvant chemotherapy). We thus considered eRPV as a surrogate of RPV and subsequently used eRPV in a subset of the TCGA dataset without publicly available CT scans, as an extension of RPV (Noted as "eRPV" in Fig. 3d, e, Supplementary Figure 7a, 7c, 7e and h-j).

Overall, we observed RPV to be associated with OS, independent of known clinical prognostic factors, suggesting that it may reflect distinct aspects of clinically relevant variation across HGSOC.

**Biological interpretation of the radiomic prognostic vector**. To understand tumor biological characteristics linked to RPV, we evaluated enrichments of Kyoto Encyclopedia of Genes and Genomes (KEGG) pathways from Spearman correlation coefficients of gene expression with RPV (Fig. 3a, b; false discovery rate (FDR) < 0.05); the full lists of pathways are given in Supplementary Data 3 and 4.

We found that ECM−receptor interaction and focal adhesion were the two pathways most significantly enriched for associations with high RPV. These two pathways contained ECM components (TIMP3 ($r = 0.530$), COL11A1 ($r = 0.460$)) and focal adhesion receptors (ITGA5 ($r = 0.368$), ITGB5 ($r = 0.387$)), and from previous studies both pathways were enriched in stroma[20,21]. Accordingly, genes with expression correlated to high RPV were significantly enriched for genes expressed in the stromal component (Fig. 3c, chi-squared test $p < 0.0001$). Additionally, RPV was positively correlated with a stroma marker, fibronectin, at the protein level in both the TCGA and the HH cohorts (Fig. 3d and Supplementary Figure 7a). Furthermore, high RPV was associated with high proportion of tumor-associated stromal cells, evidenced from both histological data (Fig. 3e) and stroma score estimated from transcriptomic data[22] (Supplementary Figure 4b). A lower tumor cell content is inversely related to high stromal content. Consistent with previous results, we noted that higher RPV was associated with lower tumor cellularity (Fig. 3e) in the TCGA cohort and the same trend was observed in the HH cohort (Supplementary Figure 7d). These associations between molecular and histological characteristics with RPV were also observed with eRPV in a subset of the TCGA dataset without publicly available CT scans (Supplementary Figure 7a, 7c and 7e).

Besides stroma-related pathways, a number of proliferation and DNA damage response (DDR) pathways, including DNA replication, cell cycle, mismatch repair, base excision repair, nucleotide excision repair and homologous recombination, were among the top pathways activated in the RPV-low tumors (Fig. 3b). To verify the validity of the pathway analysis, we analyzed reverse phase protein array (RPPA) data from both HH and TCGA cohorts and found the expression of proliferation and DDR pathway marker proteins including Stathmin 1, FoxM1 and Rad51 to be higher at the protein level in tumors with low RPV in the two independent datasets (Fig. 3d and Supplementary Figure 7a), which was consistent with our transcriptomic and pathway analysis. Existence of highly proliferative cancer cells with impaired DDR mechanism (e.g. TP53 mutation) could elicit accumulation of DNA damage[23]. Accordingly, higher tumor mutational burden and CNA burden were observed in RPV-low tumors (Supplementary Figure 7k-7l). Collectively, these molecular features suggest that RPV-low patients may benefit from DDR inhibitors (PARPi) and immunotherapy (anti-PD1/PD-L1)[24]. Potential alternative therapeutic targets based on the molecular characteristics associated with RPV are listed in Supplementary Table 8.

Molecular subtype, BRCA1/2 mutations and CCNE1 amplification are well-established molecular characteristics contributing to primary chemotherapy response and prognosis. However, they were not found correlated with RPV, highlighting the independent disease mechanisms associated with RPV (Fig. 3e; Supplementary Note 1).

Overall, stromal phenotype on one hand, and proliferation and DDR pathways on the other, were respectively activated in RPV-high and RPV-low tumors, all of which are potential actionable therapeutic targets in HGSOC.

**The reliability and reproducibility of the radiomic profile**. To determine the reliability and reproducibility of the radiomic profile, we assessed potential sources of error during radiomic data preparation. Firstly, we assessed the batch effect of CT scanner types on radiomic profile and RPV. Principal component analysis of the radiomic profile for all tumors showed no association at all with the five vendors or two types of matrix and was only moderately

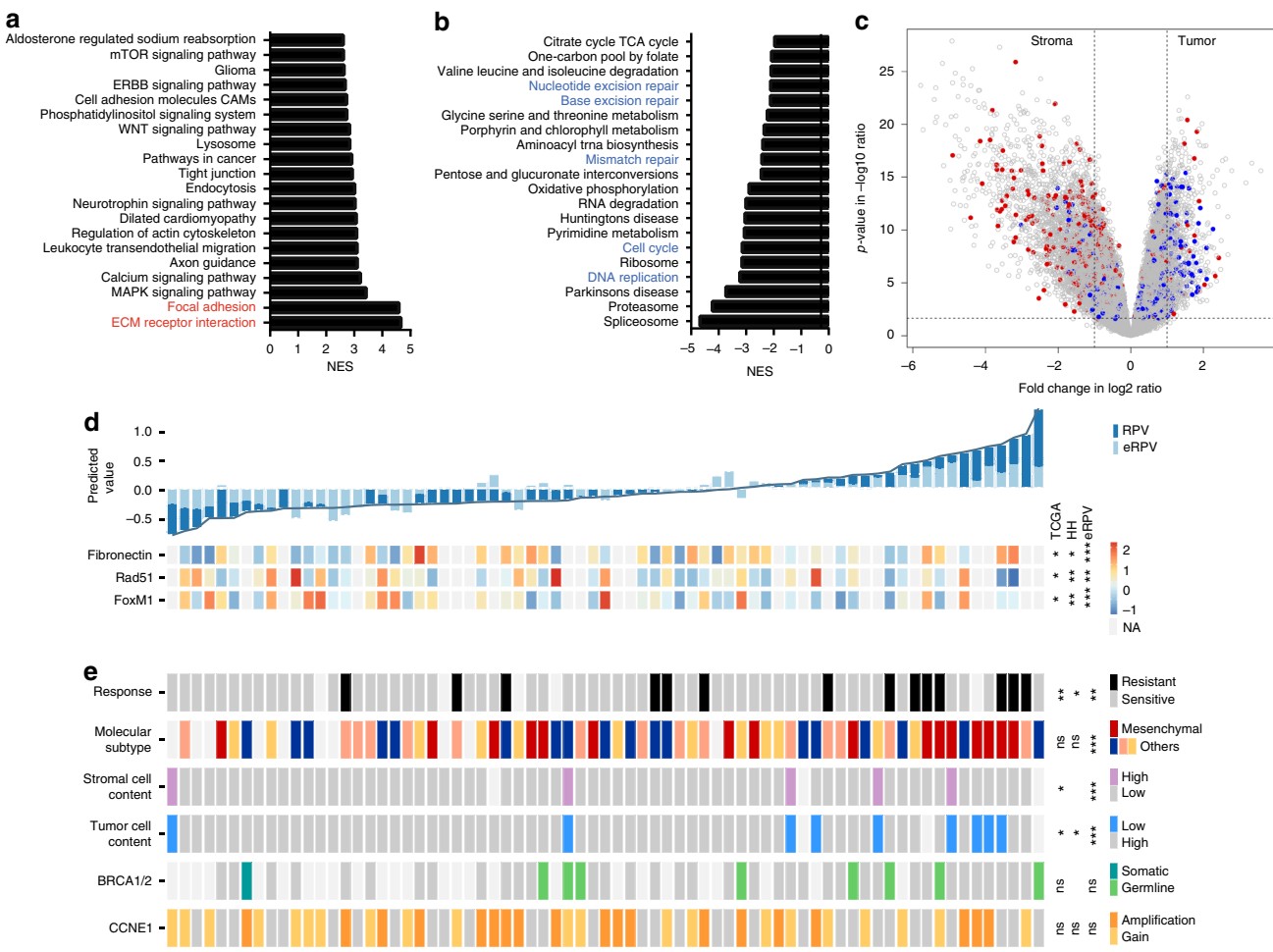

**Fig. 3** Molecular characteristics associated with RPV in HGSOC. Gene set enrichment analysis identified **a** RPV-positively correlated biological pathways and **b** RPV-negatively correlated biological pathways from KEGG pathway database (FDR < 0.05). NES normalized enrichment score. **c** Volcano plot showing the differential expressed genes between stroma and tumor epithelial component from a public dataset, GSE40595. The genes that positively correlated with RPV are highlighted in red (r > 0.3, Spearman correlation); genes that are negatively correlated with RPV are highlighted in blue (r < −0.3, Spearman correlation). **d** Heatmap showing correlation of protein expression (Fibronectin, Rad51 and FoxM1) with RPV for 47 cases in the TCGA validation dataset. The significance between these protein features with RPV was indicated for 119 cases in the HH cohort and with eRPV from 353 additional TCGA cases. Top panel, RPV ranked from low to high (left to right) and their corresponding eRPV (light blue). Lower panel, protein expression level of Fibronectin, Rad51 and FoxM1. p values are given by one-sided Spearman's correlation test as validation of the transcriptomic analyses. **e** Clinical, histological and genetic characteristics associated with RPV in the TCGA and HH cohorts. Each rectangle block represents one patient in the TCGA validation dataset. The significance of association between these characteristics with RPV in the TCGA validation dataset, HH cohort and eRPV in additional TCGA dataset is indicated on the right side. The significance is indicated on the right from Kruskal−Wallis test (molecular subtype) or two-tailed Wilcoxon rank-sum test (others). The association between RPV and stromal component is shown in (**a**), (**c**), (**d**) and (**e**); The association between RPV and proliferation or DNA damage response is highlighted in (**b**) and (**d**). \*\*\*p < 0.001, \*\*p < 0.01, \*p < 0.05, ns p > 0.1. RPV radiomic prognostic vector, HGSOC high-grade serous ovarian cancer, TCGA the Cancer Genome Atlas, KEGG Kyoto Encyclopedia of Genes and Genomes

associated with the scan thickness (Fig. 4a and Supplementary Figure 12a-12c; Range: 1–10 mm). Thus, RPV is deemed to be unaffected by the types of CT scanner investigated.

To assess the reliability of radiomic data generated, we investigated the feature-wise correlation in HH and TCGA cohorts (Fig. 4b). A consistent feature-wise correlation across independent studies is an indicator of high reliability. The feature-wise correlation in HH and TCGA cohorts were strongly correlated (r = 0.817, p < 0.0001, Pearson correlation), signifying a relatively consistent structure within the radiomic profile compared with molecular profiles from RPPA (Supplementary Figure 12c).

In the present study, the primary tumors from CT scans were initially segmented by radiologists, then analyzed by the TexLab 2.0 software. The segmentation process could potentially cause

interobserver errors due to the manual nature of the procedure; therefore, we investigated the effect of eight deformations (from −4 to +4 voxels) of the original segmentations on RPV (Fig. 4c). The difference between the deformed and original RPV are shown for each deformation from 106 scans in Fig. 4d. We found that erosion of the segmentation generally amplified the original RPV and dilation had an opposite effect, which resulted in an inverse correlation between the difference in RPV and increase of voxels. Importantly, the variation in RPV was unremarkable within the range of 1-voxel erosion (mean difference: 0.105, sd: 0.167) and 3-voxel dilation (mean difference: −0.0418, sd: 0.125). The interobserver variation in RPV, determined from segmentation made by two independent radiologists for 21 scans, fitted well within this range (Supplementary Figure 12e).

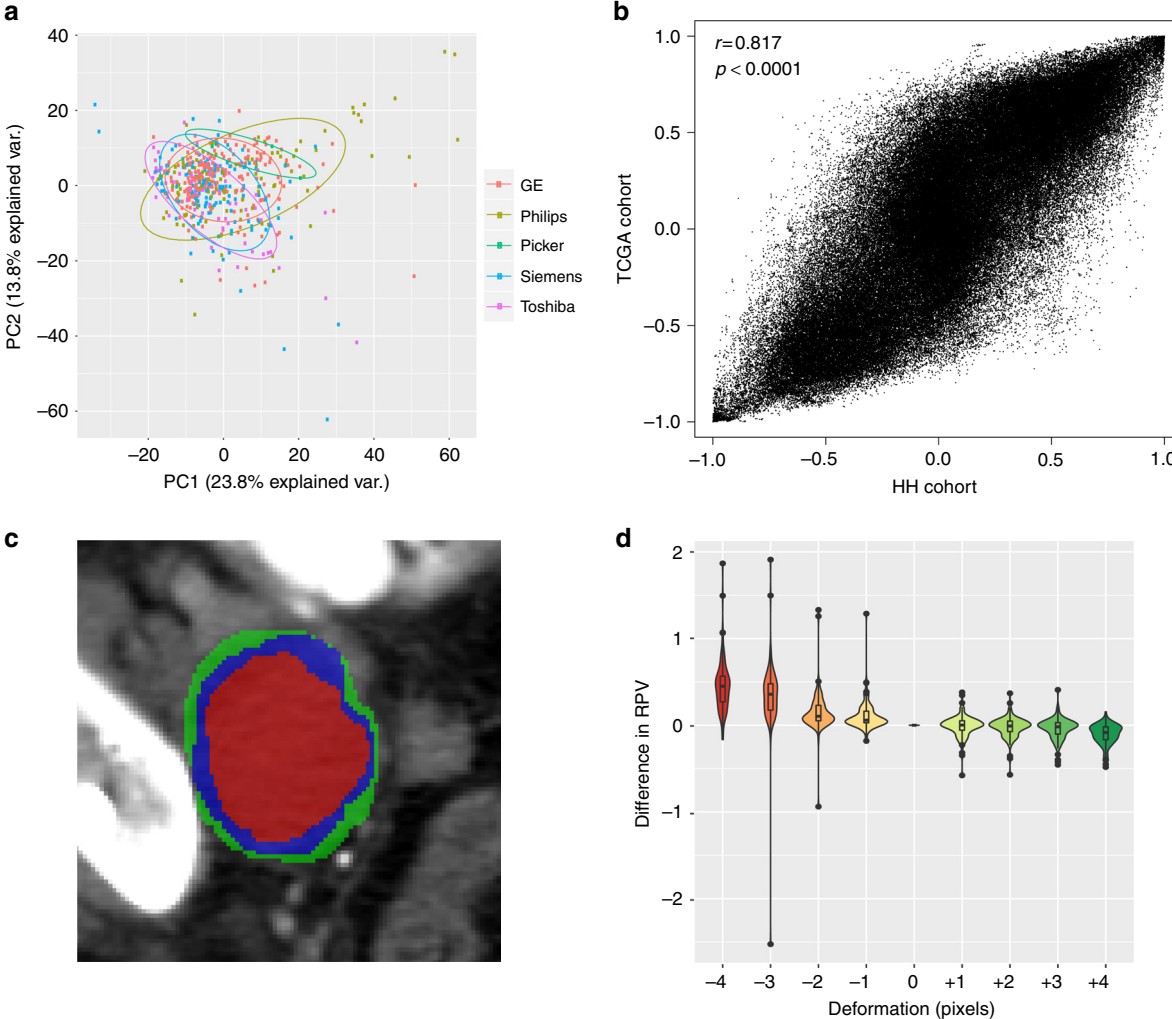

**Fig. 4** The reliability and reproducibility of radiomic profile. **a** Principal component analysis (PCA) plot of radiomic profile by scanner manufacturers. **b** Radiomic feature-wise correlations of two independent cohorts. $x$-axis indicates the correlation between any two radiomic features in the HH cohort, $y$-axis indicates the corresponding feature-wise correlation in the TCGA cohort. The Pearson correlation coefficient and $p$ value is indicated. **c** An illustration of erosion (red) and dilation (green) on the original segmented CT scan (blue). **d** Impact of segmentation deformations on RPV. $x$-axis indicates the range of deformations from erosion by 4 voxels ($-4$) to dilation by 4 voxels ($+4$) and 0 is the original segmentation. $y$-axis indicates the difference between deformed RPV and original RPV. HH Hammersmith Hospital, TCGA the Cancer Genome Atlas, CT computed tomography, RPV radiomic prognostic vector

## Discussion

In the present study, we obtained and analyzed a comprehensive radiomic profile containing 657 features for 364 EOC cases in total—the largest study of its kind for EOC—and we discovered a novel radiomics-based prognostic signature, RPV, that not only has strong prognostic power (HR > 3), but is also noninvasive and readily accessible, compared to the existing molecular profiles and clinical factors deemed prognostically relevant[12,14]. In contrast to previous studies that lacked interpretation of the prognostic signature, we comprehensively profiled biological and clinical features associated with RPV that will help guide future clinical decision processes in a reliable and reproducible fashion.

Several previous studies have attempted to develop predictive and prognostic tools based on molecular profiles from tumor biopsies such as gene expression, DNA methylation, CNA, and more recently microRNA and circulating tumor DNA[12,14,15,25–27]. These molecular prognostic models are challenging to translate into routine clinical use due to the invasiveness of a biopsy, insufficient prognostic power due to the vast intratumor heterogeneity, high assay costs, and most importantly, the significant time constraints that are associated with the molecular assay

procedures. The prognostic model we propose is simple, built solely on the information extracted from a patient's routine preoperative CT scan at the presentation of the disease and hence readily accessible without additional costs or time delays, knowing that majority of the HGSOC patients will have CT scans prior to the treatment (compared to PET, MRI or ultrasound). The entire primary ovarian mass is segmented, signifying that any prognostic or biological information extracted is more representative of the disease compared to a single site biopsy. Moreover, RPV is stable across the CT scanner types and the segmentation process, thus limiting the number of potential restrictions for clinical exploitation in the future. We have constructed a software pipeline which is able to compute the RPV of 80 EOC datasets within 5 min on a standard computer. Beyond RPV, the dataset could be mined in a supervised manner for new gene- or protein-radiomics interactions.

We employed manually engineered features as the main component of radiomic analyses; this approach is backed by the current literature in the field[1]. While some studies[28] have started to investigate the application of deep learning for radiomics via the extraction of thousands of deep features from convolutional

networks, the small sample size coupled with the curse of dimensionality in machine learning pose a hurdle for proper evaluation of deep learning features for radiomic analysis. The availability of thousands of annotated or segmented 3D medical datasets would allow for a more robust evaluation and open the possibility for applying transfer learning on 3D medical images, as is currently done with 2D images[29–31].

RPV consists of four radiomic features: (a) FD_max_25HUgl (coefficient: −0.0876), (b) GLRLM_SRLGLE_LLL_25HUgl (coefficient: 0.0869), (c) NGTDM_Contra_HLL_25HUgl (coefficient: 0.165), and (d) FOS_Imedian_LHH (coefficient: 0.250). All the features appear to have approximately even weighting and relate to tumor macro-architecture at the 25 Hounsfield Unit gray level (and discrete wavelet filters). In biological terms, the individual components of RPV combine to define the tumor-mesoscopic structure: (a) maximal fractal dimension of the tumor and its microenvironment, which was negatively correlated with survival, together with the following positively correlated features; (b) proportions of runs that have short lengths in the low pass filtered image; a function which gives coarse low-density textures, e.g. intermixed fibrotic stroma and tumor cells; (c) the average visual contrast across the tumor weighted by sharpening in the $x$-axis, and blurring in the $y$ and $z$ axes reflecting local heterogeneity, and (d) the median of the distribution of voxel intensities across the entire tumor weighted by blurring in the $x$-axis and sharpening in the $y$ and $z$ axes, reflecting global heterogeneity, respectively. A visual representation of the four radiomic features is shown in Supplementary Figure 13.

In addition to building a prognostic model, we further demonstrated that the radiomics-derived signature is closely linked to a stromal phenotype and DNA-damage response through genetic, transcriptomic, proteomic and histological analysis. This finding is consistent with the poor prognostic value of stromal phenotype identified in many cancers including ovarian[32–36], pancreatic[37], prostate[38], colorectal[35,39], gastric[32], lung[35] and breast cancer[40]. Tumor stroma consists of immune cells, endothelial cells, fibroblasts and extracellular-matrix (ECM)[41] all of which could directly contribute to outcome via distinct mechanisms in EOC[42–45]. We demonstrate, based on the strong association between RPV and response to primary chemotherapy or surgery, that patients with high RPV have a significantly high risk of failing quality surgery or systemic strategies and suggest that they possibly need to be directed towards alternative therapeutic approaches including stroma modifying therapies (e.g. ClinicalTrials.gov ID: NCT03363867).

Interestingly, in our HGSOC cohort we did not observe a strong association between RPV and any single cancer driver events including ovarian cancer "molecular subtypes", specific gene mutations or CNA, suggesting that the RPV phenotype and related poor prognosis may be shaped by noncanonical genetic alterations or pathways.

There are some limitations of the present study: firstly, the study had a retrospective design albeit with two independent validation datasets. A future prospective study or analysis of retrospective randomized clinical trial data is required to validate RPV in a more general HGSOC population. Secondly, as the stromal component contains a mixture of cells of different origins and ECM composition, the exact elements in the stroma measured by RPV remain unclear. A study to associate RPV with each component in stroma including fibroblast activation, immune cell infiltration and ECM density is necessary to better understand the basis of the prognostic power of RPV. In addition, EOC patients often present with bilateral disease and one tumor was chosen to represent the patient in this study. Further investigation into heterogeneity in RPV for bilateral tumors may further help optimize the prognostic model.

In summary, we have discovered and validated a novel mathematical descriptor of tumor phenotype and prognosis that convincingly fulfills an unmet need in the management of patients with EOC, and have demonstrated a disruptive technology that opens the way for multiple classifications of patients and rapid patient entry into clinical trials at the point of care.

## Methods

**Patient cohort and biospecimen collection**. This is an observational study of patient data (including data related to fresh frozen tissue, imaging and clinical annotations) from the Hammersmith Hospital (HH), Imperial College Healthcare NHS Trust and from the TCGA study. All procedures involving human participants were done in accordance with the ethical standards of the institutional and/or national research committee and with the principles of the 1964 Declaration of Helsinki and its later amendments or comparable ethical standards. Ethical approval for retrospective analysis of human data was obtained under the Hammersmith and Queen Charlotte's & Chelsea Research Ethics Committee approval 05/QO406/178 and informed consent was waived, typical for retrospective analysis of anonymized imaging data.

EOC patients included in the Hammersmith cohort were treated at the Hammersmith Hospital (HH), Imperial College London NHS Trust between June 2004 and November 2015. The patients were identified based on the availability of fresh frozen tumor tissue samples and preoperative CT images.

Patient demographics, surgical and tumor related data were collected retrospectively from medical records and the multidisciplinary team (MDT) notes by the clinical members of the team are summarized in Supplementary Table 1. PFS and OS were defined as the time from the date of surgery until the date of first relapse or death, respectively. Staging was defined according to FIGO-criteria for ovarian epithelial carcinoma. Optimal debulking was defined by postoperative residual disease < 10 mm since this criterion was applied to majority of the retrospective patients. Primary chemotherapy resistance was defined as stable disease, a partial response or progressive disease during the first-line chemotherapy.

Tumor cellularity was quantified from hematoxylin and eosin-stained sections by an experienced pathologist. Based on the multidimensional scaling analysis we performed on the RPPA data, only samples with more than 30% tumor cellularity were included in the RPPA analysis.

A subset of EOC patients from TCGA study were used as the validation cohort. The preoperative CT images for these cases were downloaded from the cancer imaging archive[46] (http://www.cancerimagingarchive.net/). This was a multicenter cohort with patients originating from Memorial Sloan Kettering (30 cases), Mayo Clinic—Rochester (4 cases), University of Pittsburgh (10 cases), UCSF (16 cases) and Washington University (9 cases). The clinical and histological data were downloaded from UCSC cancer browser (https://genome-cancer.ucsc.edu/).

**Clinical and surgical pathways**. The management of all patients and the indications for surgery were discussed within a multidisciplinary team as per the UK National Health Service (NHS) guidelines. All operations were performed through a midline laparotomy by a specialized dedicated multidisciplinary team within a maximal effort approach aiming to achieve total macroscopic tumor clearance. Standard surgical procedures included peritoneal cytology, extrafascial hysterectomy, bilateral salpingoophorectomy and infra-gastric omentectomy. When indicated, additional procedures, such as dissection of macroscopically suspicious pelvic and paraaortic lymph nodes, bowel resection, splenectomy, diaphragmatic stripping/resection and/or partial resection of other affected organs (e.g. urinary bladder, liver/liver capsule, pancreas, lesser sack) were performed in order to achieve optimal tumor debulking. No systematic pelvic and paraaortic lymph node dissection was performed routinely in the absence of suspicious bulky lymph nodes (<1 cm).

Ninety-seven percent of patients were treated with a platinum-based chemotherapy mainly in a combination regimen with paclitaxel or as monotherapy in isolated cases.

**Clinical follow-up of patients**. Patients were regularly evaluated at the end of their treatment for evidence of disease recurrence. Clinical examination and CA-125 assessment (if the preoperative value was elevated) were performed every 3 months for the first 2 years and then 6-monthly. A CT/MRI-scan was ordered if the above examinations revealed any pathology. An isolated CA-125 increase was not regarded as a recurrence.

**CT segmentation and radiomic analysis**. As patients were referred to the cancer center from a network of cancer units, contrast-enhanced CT scans were acquired at multiple institutions using different manufacturers and different imaging protocols.

For both the HH and the TCGA datasets, the primary tumor masses were segmented separately by experienced radiologists (M.A., G.A.) using ITK snap (Version 3.2, 2015) and then all segmentations were checked in consensus with a radiologist with over 16 years' experience of ovarian cancer imaging (A.R.). We included the entire primary tubo-ovarian mass (cystic and solid components). If both

adnexae were involved, then both were included in the analysis, either as two separate segmentations or as a single segmentation if the mass was confluent. We segmented the entire primary mass including cystic and solid components, but excluded ascites. The segmentations only included tissue that was considered highly likely to be cancer by the expert reader. Areas of doubt on CT were not included in any segmentations. Inter-observer variation was also measured by comparing independent segmentation from two radiologists using the TCGA cohort.

For this study, the primary tumor mass segmentations were used as input for the in-house texture analysis software package (TextLAB 2.0) developed in MATLAB 2015b (Mathworks Inc., Nathick, Massachusetts, USA)[16].

Using methodologies for feature extractions[1,47–53], we defined 657 radiomic image features that describe tumor characteristics. The features can be divided into several groups: 1. Shape and Size features; 2. First-order statistics; 3. Second-order statistics; 4. Wavelet features.

The first group relates to statistics based on the shape of the tumor, e.g. compactness or sphericity. The second group quantified tumor voxel intensity characteristics. Group 3 consists of textural features that quantify different measures of three-dimensional intratumoral heterogeneity. The wavelet features group calculates the features in groups 2 and 3 after performing wavelet decompositions of the original image using high-pass or low-pass filters from the coiflet 1 family of wavelets. All feature algorithms were implemented within MATLAB.

**Transcriptomic, proteomic and copy-number analysis.** Frozen tumor tissue pieces ($n = 314$) were placed into ceramic bead tubes (Stretton Scientific) for protein extraction by the Functional Proteomics RPPA Core Facility, MD Anderson, USA. Protein concentration was determined following extraction and adjusted to 1.5 μg/μl. Proteins were denatured by 1% SDS plus beta-Mercaptoethanol and serially diluted for subsequent Reverse Phase Protein arrays.

For each tumor in the study, one frozen tumor piece was placed into a tube containing 500 μl RLT buffer from RNeasy kit (QIAGEN) and one Retsch 6 mm steel core bead. Tubes were placed into well adapters of a Tissuelyser II (QIAGEN) and tissues were lysed at 15 Hz for 2 min. Tubes were centrifuged briefly and 320 μl was removed for subsequent RNA extraction using the RNeasy kit (QIAGEN) according to the manufacturer's instructions. RNA concentrations were quantified using the Bioanalyzer system (Agilent).

For DNA extraction, 450 μl of Buffer ATL from the QIAAMP DNA kit (QIAGEN) was added to the centrifuge tube, and DNA was extracted following the manufacturer's instructions and quantified using QuBit (Thermo Fisher Scientific).

RPPA arrays were carried out and analyzed by MD Anderson Cancer Center[54]. Briefly, protein lysates were diluted and loaded onto nitrocellulose-coated slides that had been preconjugated with primary antibodies. Each protein was then visualized via a colorimetric reaction and quantified by Array-Pro Analyzer. The raw expression values were then normalized to protein loading and quantified by means of standard curves. Log2 transformed and median-centered data were used for the downstream analyses.

To perform molecular subtyping, total RNA from each individual case was reverse transcribed into cDNA, followed by amplification with a pool of indexed primers that target a predefined gene list (42 genes)[13]. The primers were selected from the Illumina DesignStudios. The cleaned PCR product underwent QC by Tapestation (Agilent) to confirm the amplicon sizes. Forty-eight samples were multiplexed in one single MiSeq run. SR 50 bp were used to generate approximately 20 million reads per run.

Copy-number estimates for *CCNE1* in 131 tumor samples from the HH cohort were obtained through quantitative qPCR[10]. ΔCt values for tumor samples (*CCNE1* relative to the endogenous control *LINE1*) were normalized to equivalent ΔCt values from reference (normal Fallopian tube cell line DNA) with an assumed *CCNE1* copy number of 2.

**Unsupervised clustering and signature discovery.** A simple spectral cluster analysis was performed using radiomic data from patients with both radiomic and genomic copy-number data available. First a similarity measure was computed for each patient with radiomic and CNA profiles as the average Spearman correlation coefficient. The profiles of pair-wise similarity were then used to compute the Euclidean distance between each pair of patients. Visual inspection of a hierarchical clustering dendrogram was used to select three clusters of patients, so that patients from a given cluster tended to share correlated radiomic profiles.

The number of genes affected by CNA was calculated for each tumor sample, so that the distribution of the logarithm of these numbers could be compared for tumors from patients belonging to different clusters.

Kaplan−Meier curves were drawn for PFS and OS using the "survfit" function from the "survival" package in R. The statistical significance of the difference in these survival measures across the three patient clusters was calculated using the log-rank test implemented in the "survdiff" function.

Unsupervised hierarchical clustering of radiomic profiles were performed using hclust and cutree function in R 3.3.1. The raw radiomic data were firstly scaled by mean and centered. Pearson correlation-based distance and complete linkage was used to obtain the final clusters indicated in Fig. 1e. The resulting clusters were confirmed by repeating the clustering analysis using Euclidean's distance. The optimization of the radiomic clusters is indicated in Supplementary Figure 14. The heatmap was generated using the "heatmap.plus" package in R 3.3.1.

Least absolute shrinkage and selection operator (LASSO) analysis was performed to build a prognostic model for OS using radiomic data. We first selected for the discovery dataset (HH discovery) a group of HGSOC patients who had primary debulking surgery as well as patients not in the unsupervised subgroup 2 (which had different slice thickness compared to other subgroups, Supplementary Figure 14). All the other HGSOC patients were used as the HH validation dataset and the HGSOC patients in the TCGA cohort were used as the TCGA validation dataset. We selected a large pilot dataset for discovery (HH discovery, $n = 136$) and the number of patients in the two validation datasets fulfilled sample size estimate (73 cases needed after accepting the alpha of 0.05 and beta of 0.25. HR of 2.78, 31.6% cases in the high-risk group, median survival of 5 years in the low-risk group and median follow-up as 5 years). To generate a prognostic model of OS, a univariate Cox regression was performed between individual radiomic features and OS, which was adjusted for stage, slice thickness and residual disease in the HH discovery dataset. Since it was not possible to decide the more prognosis-related tumor for bilateral tumors and we had demonstrated close similarity between the two tumors, we included both bilateral tumors at the model-building stage. The radiomic features with FDR < 5% were selected as input for LASSO regression using glmnet package in R 3.3.1. "Cox" was set as the family in the model. Ten-fold cross-validation was performed using cv.glmnet function to select lambda minimum to give the minimum cross-validated error. The resulting four radiomic features with coefficients were used to calculate a predictive index—RPV—for each patient. The RPV was used to perform subsequent continuous Cox regression and Kaplan−Meier analysis with OS and PFS. For patients with bilateral tumors, the tumor that gave the higher RPV was selected since it resulted in better performance compared with the one with lower RPV. After considering the distribution of RPV and number of patients in each subgroup, $K$-means clustering was applied to split the patients into three subgroups (low risk: min−0.0950, medium risk: 0.0950–0.658, high risk: 0.658−max). The same criteria were used to obtain subgroups in the validation cohorts. For validation, the radiomic data from the TCGA dataset and the HH validation dataset were initially scaled and centered. The RPV was calculated using the four radiomic features with coefficients derived from the discovery set. For those cases with bilateral tumors which resulted in two RPV values, the higher RPV was selected for the survival analysis. For multiple Cox regression of RPV, the slice thickness was introduced as additional variable. Only cases with complete clinical information (stage, age and postoperative residual disease) and slice thickness were included in the multivariable Cox regression analysis. REMARK guidelines were followed when reporting RPV as a prognostic marker in this study[55].

Similar procedure was applied to generate eRPV with some modifications. Gene expression profile from Affymetrix HT Human Genome U133a (Level 2) and Agilent 244K custom gene expression G4502A_07_3 (Level 3) from the TCGA study were downloaded from UCSC cancer browser (https://genome-cancer.ucsc.edu/) and TCGA data portal (https://cancergenome.nih.gov/). Spearman correlation was applied to obtain a list of genes correlated with RPV (FDR < 0.25 for Affymetrix and FDR < 0.1 for Agilent). The gene list obtained was used to perform feature selection and linear regression with RPV using "glmnet" package. "gaussian" was set as the family in the model and a tenfold cross-validation was applied. The resulting weighted gene lists contributing to eRPV are given in Supplementary Data 5 and 6.

Gene set enrichment analysis (GSEA) was performed for RPV-correlated genes in the TCGA dataset. The Level 3 RNA-sequencing dataset of EOC from the TCGA project was downloaded from UCSC cancer browser (https://genome-cancer.ucsc.edu/) and the gene-level transcription estimates were obtained in reads per kilobase million. Spearman correlation coefficient was determined from RPV and all the genes respectively. The full list of correlation coefficients was used as the pre-ranked list in GSEA 2.1.0 with KEGG database, 1000 of permutations and classic enrichment statistic.

Differential gene expression between tumor and stroma in HGSOC was analyzed using limma package from Bioconductor in R 3.3.1. The Robust Multi-array Average (RMA) normalized microarray dataset (GSE40595) was downloaded from GEO (https://www.ncbi.nlm.nih.gov/geo/). The empirical Bayes moderated $t$-statistics were computed comparing gene expression from HGSOC stroma and tumor epithelial component. The $p$ value derived was adjusted for multiple testing using Benjamini–Hochberg procedure.

**Statistical analysis.** Standard statistical analysis was applied to all the figures as appropriate and indicated in the figure legends. All samples were used once. Multiple testing was corrected with FDR method. All the statistical analyses were conducted in R 3.3.1.

**Reporting summary.** Further information on experimental design is available in the Nature Research Reporting Summary linked to this article.

**Code availability.** The R script that was used to reproduce the key findings and generate figures are publically accessible in Mendeley Data with the identifier https://doi.org/10.17632/4c5znk5m8t.1.

# Data availability
The radiomics, clinical, RNA-sequencing and proteomics data generated in this study have been deposited into the Mendeley database under the accession code:

https://doi.org/10.17632/4c5znk5m8t.2. The gene expression, copy number alteration and RPPA data from the TCGA project[56] were downloaded from the UCSC cancer browser (https://genome-cancer.ucsc.edu/). The gene expression microarray data from the Tothill dataset and laser capture microdissected ovarian tumor tissue were downloaded from the NCBI Gene Expression Omnibus with accession numbers GSE9891[14] and GSE40595[57]. The CT scan data from the TCGA ovarian cancer project were downloaded from the Cancer Imaging Archive[46] (http://www.cancerimagingarchive.net/). All the other data supporting the findings of this study are available within the article and its supplementary information files and from the corresponding author upon reasonable request.

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

## Acknowledgements

This article is an independent research funded by the National Institute for Health Research (NIHR), Imperial Biomedical Research Centre (BRC), the Imperial College Experimental Cancer Medicine Centre (ECMC) and Imperial College London Tissue Bank. The views expressed are those of the author(s) and not necessarily those of the NHS, the NIHR or the Department of Health. The authors acknowledge Dr. Mona El-Bahrawy and Dr. Roberto Dina for histology expertise, Dr. Bill Crum for computational support and Prof. Gordon Mills of MD Anderson Cancer Centre for RPPA support. E.O.A. acknowledges programmatic support from the UK Medical Research Council (MC-A652-5PY80) and Cancer Research UK (C2536/A16584). H.G., C.F., H.L. and P.C. acknowledge support from Ovarian Cancer Action.

## Author contributions

E.O.A., A.R., H.G. and C.F. conceived and designed the project. A.T. developed TexLab 2.0. M.A., G.A., J.L. and A.R. collected and segmented the radiological data. A.T., K.N., S.T.W., M.A.H., H.L. and C.F. collected the clinical data. H.L., P.C. and J.L. collected the RNA-sequencing, RPPA and qPCR data. H.L., E.C. and F.K. provided bioinformatics and computational analysis. E.O.A., H.L., M.A., A.T., P.C., E.C., D.D.L.B., H.G., C.F. and A.R. contributed to the interpretation of data. H.L. and E.O.A. wrote the manuscript. All authors edited the manuscript.

## Additional information

**Competing interests:** H.G. is an employee of AstraZeneca. The other authors declare no competing interests.

