## [Peer Review File · Nature Communications]

Reviewers' Comments:

Reviewer #1:

Remarks to the Author:

A Mathematical descriptor of tumor mesoscopic structure...

Lu et al., Nat Communications

Summary

The authors have discovered and validated imaging characteristics of ovarian cancers taken from CT scans pre-operatively, which have prognostic value. They associate these findings with stromal biological factors.

Major Issues

1. Any prognostic analysis of ovarian cancer should include performance status. This isn't mentioned in the manuscript. It should be included in the authors' univariate and multi-variate analyses.
2. The HH validation patient group were split approximately 50:50 (Table S2) into patients who had surgery first or chemotherapy first. The HH discovery and TCGA data were based on patients who had surgery first. This is important because (a) it is unclear whether the scans from the HH validation set, upon which the RPV was calculated, would have been taken after 3-4 cycles of chemotherapy, which might/might not impact on imaging characteristics. This is of interest to readers in any case. Nearly all patients treated with pre-operative chemotherapy would have had CT scans before neoadjuvant chemotherapy and pre-operatively. Could the authors comment on which CT scan they evaluated in patients treated with neoadjuvant chemotherapy and what changes/ stability of RPV were seen over time?
3. As a clinician I would really like to know whether RPV predicts suboptimal resection. At present the paper identifies a prognostic factor, which might be associated with stromal biology. However, in terms of clinical decision making, we need an imaging parameter that would predict disease that cannot be completely resected at initial surgery, which would lead MDTs to recommend initial chemotherapy rather than surgery. Although there were some comments that refer to this question, I couldn't see it directly addressed. Could the authors comment?
4. Lines 96-98: The comment on the similarity between two ovarian tumours in the same patient is really important. Could the authors comment on the similarity between ovarian cancer deposits and disease in the omentum and peritoneum i.e. the other anatomically important areas in the patient's body? Do all deposits in a patient resemble each other?
5. Line 136 onwards: It is not clear to me whether the TCGA data set were evaluated through a surrogate of RPV, called eRPV, which is derived through a comparison of gene expression profiling and radiomics in the Tothill and TCGA data sets. Line 143 suggests that no TCGA scan data were used but instead molecular surrogates were used instead. Thus, arguably the TCGA data aren't true validation cohorts. If I have understood this point correctly, then I think the authors have been open about it but the point should be included in the discussion as a limitation of the study. In particular the text on lines 160-1 then needs clarification about whether the RPV or eRPV is used when commenting on fibronectin in the TCGA cohorts.
6. Patients included in this study period would have been those eligible for treatment with bevacizumab, an anti-angiogenic and therefore stroma-modifying drug. Did the authors have the opportunity to look at the impact of bevacizumab in this population?
7. Table 2 is of concern: Nearly all ovarian cancer series identify age, stage, residual disease and performance status as independent prognostic variables. Can the authors explain why age is not statistically significant in any of the data sets; why residual disease is only significant in univariate analysis in the HH discovery set and why stage is only significant in the HH discovery set?

Minor Issues

1. In the abstract the statement that "RPV reliably identified the 5% of patients who did not survive beyond 5 years" doesn't seem correct particularly in view of the first line of the abstract. Could the authors check the 5% figure?

2. The images/tables are not listed in cited order in the manuscript.
3. Ovarian cancer is cystic and associated with ascites. For the clinical audience please explain how such areas on the scan were treated i.e. were they eliminated from the analysis?
4. Figures 3 and S4 were very difficult to understand from the figures and legend.
5. Figure S9 doesn't explain to the general scientific reader what these parameters mean visually. Could the authors explain what they mean by wavelets?
6. Table S5 would benefit from a significant reduction in decimal places.

Reviewer #2:

Remarks to the Author:

The authors present a CT image analysis procedure to correlated with poor survival. The analysis procedure comprises the manual definition of 657 image descriptors for association and is demonstrated for a patient cohort of epithelial ovarian cancer patients with survival information.

The authors derive a small set of descriptors that are predictive for survival and achieve a good stratification of the cohort with respect to survival. The authors corroborate the validity of this set, by identifying associations with molecular signatures independently associated with survival.

The authors claim that this set is novel, and attractive due to its non-invasive accessibility. The authors don't review other approaches to stratify patients using CT, and other radiology approaches. What would the added value of their signature be? How does stratification performance to such approaches compare? CT based approaches could be compared on the data utilized in the study. Approaches based on other imaging modalities (e.g. MRI, PET, Ultrasound) should be at least discussed.

The midterm value of this signature is questionable due to the immense advances in representation learning using deep neural networks. Such approaches circumvent the manual and biased definition of image features and learn these from data while maximizing the correlation with the supervision label (here survival). This study would be significantly strengthened with inclusion of a benchmark against such procedures. Transfer learning approaches could be used to deal with the limited number of patients (see also Thrun et al. Nature, 2017).

Reviewers' comments:

Reviewer #1 (Remarks to the Author):

A Mathematical descriptor of tumor mesoscopic structure...

Lu et al., Nat Communications

Summary

The authors have discovered and validated imaging characteristics of ovarian cancers taken from CT scans pre-operatively, which have prognostic value. They associate these findings with stromal biological factors.

Major Issues

1. Any prognostic analysis of ovarian cancer should include performance status. This isn't mentioned in the manuscript. It should be included in the authors' univariate and multi-variate analyses.

Most of the patients recruited to the study were those who could tolerate a maximal effort upfront debulking surgery, so their performance status (Eastern Cooperative Oncology Group, ECOG) were usually good enough for this highly invasive procedure. In view of the reviewer's comment, we have now collected the available ECOG status for 82 patients from both the HH and TCGA cohorts and performed univariate and multivariable analysis (Table R1). The result suggested the prognostic power of RPV is independent of ECOG status. The lack of significance of ECOG is likely due to the extremely small proportion of patients with ECOG status greater than 2 (3 out of 82 cases).

Table R1. Univariate and multivariable Cox regression analysis for overall survival in the combined HH and TCGA cohort. Only cases with available performance status were included.

HH discovery + HH validation + TCGA validation (n = 82)	Univariate		Multivariable		
	Variables	HR (95% CI)	p-value	HR (95% CI)	p-value
	RPV	3.96 (2.07-7.57)	3.22x10 ⁻⁵	4.19 (2.042-8.61)	9.44x10 ⁻⁵
	Stage	1.64 (0.560-4.80)	0.367	2.37 (0.730-7.71)	0.151
	Residual disease	3.26x10 ⁻⁹ (0-Inf)	0.998	3.57e-9 (0.00-Inf)	0.998*
	Age	0.964 (0.915-1.02)	0.167	0.966 (0.914-1.02)	0.220
	Performance status (ECOG)	0.924 (0.385-2.22)	0.859	1.50 (0.602-3.72)	0.386

As per the reviewer's suggestion, we have incorporated the table above as well as the following sentences in the manuscript.

In the results section 'Radiomic prognostic vector predicts survival' (Manuscript, Page 6):

"RPV was also found associated with OS independent of performance status in a subset of patients (Supplementary Table 6)."

2. The HH validation patient group were split approximately 50:50 (Table S2) into patients who had surgery first or chemotherapy first. The HH discovery and TCGA data were based on patients who had surgery first. This is important because (a) it is unclear whether the scans from the HH validation set, upon which the RPV was calculated, would have been taken after 3-4 cycles of chemotherapy, which might/might not impact on imaging characteristics. This is of interest to readers in any case. Nearly all patients treated with pre-operative chemotherapy would have had CT scans before neoadjuvant chemotherapy and pre-operatively. Could the authors comment on which CT scan they evaluated in patients treated with neoadjuvant chemotherapy and what changes/ stability of RPV were seen over time?

We only used CT scans prior to any treatment (chemotherapy or surgery) in this study. For the subset of patients who received neo-adjuvant chemotherapy in the HH validation dataset, we used CT scans prior to neoadjuvant chemotherapy. We agree that scans taken after neoadjuvant chemotherapy may have an impact on RPV and it will be interesting to investigate the changes and stability of RPV.

To clarify this point, we have added the following sentence as a footnote of Supplementary Table 2 in the manuscript. (Supplementary Figure, Page 21)

"Only CT scans obtained prior to neo-adjuvant chemotherapy were used for the radiomic analysis."

Please note that within the manuscript we had already accounted for the difference in treatment in this cohort of patients (neoadjuvant followed by surgery versus upfront surgery followed by chemo) by using a multivariable Cox regression analysis (Manuscript, Page 6).

3. As a clinician I would really like to know whether RPV predicts suboptimal resection. At present the paper identifies a prognostic factor, which might be associated with stromal biology. However, in terms of clinical decision making, we need an imaging parameter that would predict disease that cannot be completely resected at initial surgery, which would lead MDTs to recommend initial chemotherapy rather than surgery. Although there were some comments that refer to this question, I couldn't see it directly addressed. Could the authors comment?

As the reviewer suggested, we have tried to predict suboptimal resection with RPV. RPV was found to be higher in sub-optimal resected patients (Fig. R1a). However, this difference was too subtle to make an accurate prediction (the average area under curve (AUC) of RPV is 0.58; Fig. R1b). Although RPV predicts the patient survival to radical surgery (Supplementary Fig. 8), it does not strongly predict suboptimal resection, possibly because this is heavily influenced by surgical skills.

Figure R1. Association between RPV and sub-optimal resection. (a) High RPV was associated with sub-optimal resection in the HH and TCGA cohort. P-value is given by two-tailed Wilcoxon rank-sum test. (b) Summary of area under curve (AUC) in the validation sets after random splitting the HH and TCGA cohort with ratios of 7 (training using logistic regression) to 3 (validation) for 50 times. The average AUC is 0.58.

In a maximal effort and highly specialised setting, the proportion of patients that can be operated tumor-free is around 70%¹, and about 20% can be operated with residual disease less than 1cm. So only around 10% of patients are completely inoperable within expert teams². Usually, these patients can be preoperatively identified with conventional imaging including CT or diffusion weighted MRI. The aim of our analysis (Supplementary Fig. 8) was to reliably preoperatively identify patients who, even though could be operated tumor-free or less than 1cm residual disease, do not benefit from ultraradical surgery and thus undergo the surgical morbidity without any survival benefit.

We have included the following statement in the Supplementary Note section ‘Potential application of RPV in the surgical setting’ (Supplementary Note 1, Page 3):

“Although RPV was found associated with suboptimal resection, it did not accurately predict the resection outcome on its own, perhaps due to the influence of surgical skill on this variable (Supplementary Fig. 9).”

4. Lines 96-98: The comment on the similarity between two ovarian tumors in the same patient is really important. Could the authors comment on the similarity between ovarian cancer deposits and disease in the omentum and peritoneum i.e. the other anatomically important areas in the patient’s body? Do all deposits in a patient resemble each other?

As the reviewer suggested, we have analysed radiomic profiles from both primary tumors, disease in the omentum and peritoneum for 48 cases (Fig. R2a and R2b). By principal component analysis, two primary tumors often have similar radiomic characteristics since they are closely clustered together, which is consistent with the hierarchical clustering analysis in the manuscript. Radiomic profiles from peritoneal and omentum diseases were often found to be more distant from their corresponding

primary tumors, and clustered in a more complex way with primary tumors compared to radiomic profiles from the two primary tumors within one patient. This could be due to 1) technical factors e.g. segmentation as it is more challenging to segment peritoneal disease; 2) biological factors.

To conclude, we can see some relationships between primary tumors and disease in the omentum and peritoneum, but this is complex and currently we are limited by sample size to fully elucidate the relationship.

Figure R2. Principal component analysis of radiomic profiles from primary tumors, disease in the omentum and peritoneum for (a) 7 cases and (b) 48 cases. Each color represents a HGSOE case and each shape represents a disease location.

We have included the following paragraph in the Supplementary Note section ‘Clustering of bilateral tumors’ (Supplementary Note 1, Page 1):

“Consistently, principal component analysis of the radiomic profile from bilateral tumours of the same patient also grouped closely (Supplementary Fig. 4). In contrast, radiomic profiles from the diseases in the omentum and peritoneum were not closely grouped with their primary tumors, and clustered in a more complex way compared to the two primary tumors.”

5. Line 136 onwards: It is not clear to me whether the TCGA data set were evaluated through a surrogate of RPV, called eRPV, which is derived through a comparison of gene expression profiling and radiomics in the Tothill and TCGA data sets. Line 143 suggests that no TCGA scan data were used but instead molecular surrogates were used instead. Thus, arguably the TCGA data aren’t true validation cohorts. If I have understood this point correctly, then I think the authors have been open about it but the point should be included in the discussion as a limitation of the study. In particular the text on lines 160-1 then needs clarification about whether the RPV or eRPV is used when commenting on fibronectin in the TCGA cohorts.

The TCGA cohort (N= 582) was split into two groups: 1) 70 cases (the TCGA validation dataset) with publicly available CT scans from which we evaluated RPV; 2) The other 512 cases did not have publicly available CT scans but did have gene expression data; in this subset, we only evaluated eRPV. Therefore, the 'TCGA validation dataset' only used CT scan to derive RPV to validate the HH cohort result. Regarding to lines 160-1, fibronectin protein expression was found associated with RPV in both HH and TCGA cohort (derived from CT scans; group 1). In addition, fibronectin was also found associated with eRPV, derived from gene expression data, in the other TCGA subset (group 2).

We have modified the following sentences in the manuscript 'Radiomic prognostic vector predicts survival' to clarify this point (Manuscript, Page 7-8):

"Taking advantage of the gene expression profiles collected in parallel with radiomic profiles, we constructed a surrogate marker of RPV based on a weighted list of mRNA expressions in the TCGA validation dataset where both CT scans and gene expression profiles were available"

"We thus considered eRPV as a surrogate of RPV and subsequently used eRPV in a subset of the TCGA dataset without publicly available CT scans, as an extension of RPV (Noted as 'eRPV' in Fig. 3d-e, Supplementary Fig. 7a, 7c, 7e and h-j)."

"...These associations between molecular and histological characteristics with RPV were also observed with eRPV."

6. Patients included in this study period would have been those eligible for treatment with bevacizumab, an anti-angiogenic and therefore stroma-modifying drug. Did the authors have the opportunity to look at the impact of bevacizumab in this population?

This is a UK based study and unfortunately as per NICE guidelines, Bevacizumab is not approved for the majority of upfront operated patients; only those with residual disease more than 1cm, which was only a very small minority of our patients. We have checked all the patients records in the HH cohort and found only 3 patients treated with Bevacizumab, which is not sufficient for the analysis.

However, we plan to do a validation study in a larger European cohort, where Bevacizumab is being given to all stage III and IV patients and so we can, as per the reviewer's suggestion, evaluate the impact of Bevacizumab, especially in the RPV-high patients.

This has been added as footnote of Supplementary Table 12.

"Bevacizumab. Only 3 cases were treated with Bevacizumab in the current patient cohort."

7. Table 2 is of concern: Nearly all ovarian cancer series identify age, stage, residual disease and performance status as independent prognostic variables. Can the authors explain why age is not statistically significant in any of the data sets; why residual disease is only significant in univariate analysis in the HH discovery set and why stage is only significant in the HH discovery set?

Our patient cohort differs to other conventional ovarian cancer analyses, in that most patients were fit enough to undergo maximal effort primary/ upfront debulking surgery (86.3% of all cases), even patients of more advanced ages. Additionally, the majority of patients were able to be operated to tumor-free or with a residual disease less than 1cm, so the number of those with gross residual disease are very small (36/136 cases in HH discovery; 14/77 cases in HH validation; 15/70 in TCGA validation). Furthermore, only a small proportion of patients in our cohort had stage I-II, which is the

group that influences the prognostic value of stage (20/136 cases in HH discovery; 5/77 cases in HH validation; 13/70 in TCGA validation). Therefore, we think unbalanced groups for these prognostic variables in our patient cohort is the most likely reason for their insignificance. Were we to validate RPV in an unselected HGSOc cohort with more balanced groups, instead of a surgical one, then the significance of age, ECOG status, stage and residual disease could be apparent.

In addition to the extremely unbalanced groups, smaller sample size in each dataset (HH discovery, HH validation and TCGA validation) compared to the previous clinical trial data could contribute to the insignificance of some clinical prognostic variables. To demonstrate this point, we firstly combined all three datasets and performed Cox regression analysis for stage, residual disease, age and ECOG status. We observed an improved significance for stage, residual disease and age with larger sample size (Table R2).

Table R2. Univariate Cox regression analysis for overall survival in the combined HH and TCGA datasets.

		Univariate	
Variables		HR (95% CI)	p-value
HH discovery + HH validation + TCGA validation	Stage (n= 288)	1.75 (1.33-2.31)	7.03x10 ⁻⁵
	Residual disease (n=275)	1.64 (1.09-2.45)	0.017
	Age (n= 290)	1.01 (0.996-1.02)	0.132
	Performance status (ECOG) (n=87)	1.04 (0.454-2.36)	0.934

We next analysed these prognostic variables in the entire TCGA cohort (over 500 cases). As expected, all four variables were significantly associated with overall survival in the univariate model (Table R3). However, the significance of these variables dramatically reduced when we sampled 82 cases from the entire cohort (Fig. R3).

Table R3. Univariate and multivariable Cox regression analysis for overall survival in the entire TCGA cohort.

		Univariate		Multivariable*	
Variables		HR (95% CI)	p-value	HR (95% CI)	p-value
TCGA HGSOc cohort	Stage (n= 599)	1.57 (1.26-1.95)	6.25x10 ⁻⁵	1.10 (0.671-1.80)	0.711
	Residual disease (n=538)	1.30 (1.02-1.67)	0.035	0.956 (0.589-1.55)	0.856
	Age (n= 603)	1.02 (1.01-1.03)	0.000316	0.991 (0.988-1.03)	0.409
	Performance status (ECOG) (n=214)	1.65 (1.22-2.23)	0.00123	1.77 (1.27-2.48)	0.000818

*Only 189 HGSOc cases with complete stage, residual disease, age and ECOG information were included

Figure R3. Summary of Cox regression analysis for overall survival after sampling 82 cases for 50 times from the entire TCGA cohort. P-values of each sampling are indicated for each clinical variable in the univariate (light blue) and multivariable model (adjusting for age, stage, residual disease and ECOG status; dark blue), respectively. Dashed red line indicates $p=0.05$.

In response to the reviewer’s comment, the following statements have been added to the manuscript.

In the footnote of Supplementary Table 2 (Supplementary Figure, Page 21):

“Note that only a small proportion HGSOC patients with early stage disease or sub-optimal resection were included.”

In the results section ‘Radiomic prognostic vector predicts survival’ (Manuscript, Page 6):

“As the patients recruited to the current study are from a pro-upfront resection cohort, the lack of significance of stage, residual disease and performance status is likely due to limited cases in the low stage, sub-optimal resection, or poor performance status groups within the cohort (Supplementary Table 2).”

In the discussion section (Manuscript, Page 13):

“A future prospective study or analysis of retrospective randomised clinical trial data is required to validate RPV in a more general HGSOC population.”

Minor Issues

1. In the abstract the statement that “RPV reliably identified the 5% of patients who did not survive beyond 5 years” doesn’t seem correct particularly in view of the first line of the abstract. Could the authors check the 5% figure?

We have modified the sentence as follows (Manuscript, Page 2):

“RPV reliably identified the 5% of patients with median overall survival less than 2 years, significantly improves established prognostic methods, and was validated in two independent, multi-centre cohorts.”

2. The images/tables are not listed in cited order in the manuscript.

This issue has been addressed in the revised manuscript.

3. Ovarian cancer is cystic and associated with ascites. For the clinical audience please explain how such areas on the scan were treated i.e. were they eliminated from the analysis?

We segmented the entire primary mass including cystic and solid components, but excluded ascites. Where it was unclear if a region was part of the primary mass, we excluded this from the segmentation. Each segmentation was viewed by two radiologists to maintain the consistency.

The following sentence has been added to the Methods section (Manuscript, Page 17):

“We segmented the entire primary mass including cystic and solid components, but excluded ascites.”

4. Figures 3 and S4 were very difficult to understand from the figures and legend.

We have modified the Figure legends for these two figures:

Figure 3 (Manuscript, Page 28):

“Figure 3. Molecular characteristics associated with RPV in HGSOc. Gene set enrichment analysis identified (a) RPV-positively correlated biological pathways and (b) RPV-negatively correlated biological pathways from KEGG pathway database (FDR< 0.05). (c) Volcano plot showing the differential expressed genes between stroma and tumor epithelial component from a public dataset, GSE40595 50. The genes that positively correlated with RPV are highlighted in red ($r>0.3$, Spearman correlation); genes that are negatively correlated with RPV are highlighted in blue ($r<-0.3$, Spearman correlation). (d) Heatmap showing correlation of protein expression (Fibronectin, Rad51 and FoxM1) with RPV for 47 cases in the TCGA validation dataset. The significance between these protein features with RPV was indicated for 119 cases in the HH cohort and with eRPV from 353 additional TCGA cases. Top panel, RPV ranked from low to high (left to right) and their corresponding eRPV (light blue). Lower panel, protein expression level of Fibronectin, Rad51 and FoxM1. P-values are given by one-sided Spearman’s correlation test as validation of the transcriptomic analyses. (e) Clinical, histological and genetic characteristics associated with RPV in the TCGA and HH cohorts. Each rectangle block represents one patient in the TCGA validation dataset. The significance of association between these characteristics with RPV in the TCGA validation dataset, HH cohort and eRPV in additional TCGA dataset is indicated on the right side. The significance is indicated on the right from Kruskal-Wallis test (molecular subtype) or two-tailed Wilcoxon rank-sum test (others). The association between RPV and stromal component is shown in a, c, d and e; The association between RPV and proliferation or DNA damage response is highlighted in b and d. *** $p<0.001$, ** $p<0.01$, * $p<0.05$, ns $p>0.1$.”

Figure S4 (Supplementary Figure 10 in the revised Supplementary Figures, Page 12-13):

“Fig. S4. Construction of eRPV based on gene expression profile in the TCGA datasets (n=512) and Tothill (n=257) dataset. Selection of gene expression features to recapitulate RPV using LASSO is summarised in (a) and (b). (a) Feature coefficients were plotted against shrinkage parameter (Lambda) after performing linear regression between gene expression with RPV using LASSO in the TCGA validation dataset. (b) Selection of Lambda minimum after 10-fold cross-validation. The number of gene expression features are on the top x-axis. The Lambda minimum which resulted in the least error after cross-validation of regression between weighted expression level of 13 genes and RPV is highlighted in blue. (c) Correlation between eRPV and RPV in the TCGA validation dataset. Pearson correlation coefficient and p-value is indicated. eRPV association with OS in (d) the additional TCGA cohort without publicly available CT scans and (e) Tothill cohort. P-values are given by log-rank test. (f) Forest plot showing hazard ratio of eRPV with OS in ovarian cancer (eRPV generated from two microarray platforms) and five other cancer types in TCGA dataset. Continuous HR between eRPV and OS is given on the left and median dichotomised HR is on the right. *** $p < 0.001$, ** $p < 0.01$, * $p < 0.05$, • $p < 0.1$.”

5. Figure S9 doesn't explain to the general scientific reader what these parameters mean visually. Could the authors explain what they mean by wavelets?

We have added the following explanation in the figure legend of Fig. S9 (Supplementary Figure 13 in the revised manuscript, Page 17-18).

“Applying a wavelet filter on a CT image with a given set of filtering parameters can reveal textures with specific properties. For example, a high-pass wavelet filter reveals image areas with fine or rapid changing texture; on the other hand, a low-pass wavelet filter is the opposite of a high-pass one, and it "smooths/blurs" the image by averaging out the rapid changing texture.”

6. Table S5 would benefit from a significant reduction in decimal places.

Table S4 and S5 has been modified accordingly in the revised manuscript.

Reviewer #2 (Remarks to the Author):

The authors present a CT image analysis procedure to correlated with poor survival. The analysis procedure comprises the manual definition of 657 image descriptors for association and is demonstrated for a patient cohort of epithelial ovarian cancer patients with survival information.

The authors derive a small set of descriptors that are predictive for survival and achieve a good stratification of the cohort with respect to survival. The authors corroborate the validity of this set, by identifying associations with molecular signatures independently associated with survival.

The authors claim that this set is novel, and attractive due to its non-invasive accessibility. The authors don't review other approaches to stratify patients using CT, and other radiology approaches. What would the added value of their signature be? How does stratification performance to such approaches compare? CT based approaches could be compared on the data utilized in the study. Approaches based on other imaging modalities (e.g. MRI, PET, Ultrasound) should be at least discussed.

We agree that a new prognostic signature should be tested against the existing prognostic approaches to demonstrate its potential in clinical practice. A CT scan is usually used as part of diagnosis for high grade serous ovarian cancer (HGSOC) patients. However, there is currently no CT-based approaches to stratify HGSOC patient in clinical practice. A recent study discovered prognostic potential of a few CT-based morphological features³ (Number of locations with peritoneal disease; Peritoneal disease in paracolic gutters; Peritoneal disease around liver/right upper quadrant; supradiaphragmatic adenopathy) in HGSOC. As the reviewer suggested, we collected these CT-based morphological features for 137 cases in the HH cohort and compared these with our radiomics-based signature (RPV) (Table R4 and Fig. R4). We firstly validated the prognostic value of all four previously described radiology features (Table R4). In the multivariable Cox regression model, we found that RPV associated with overall survival independent of the radiology features, and RPV could work in synergy with other CT-based morphological features (Peritoneal disease in paracolic gutters and supradiaphragmatic adenopathy) to predict overall survival. Furthermore, we did not observe any strong correlation between RPV with any of the prognostically-relevant CT-based morphological features (Fig. R4).

Table R4. Univariate and multivariable Cox regression analysis for overall survival in the HH cohort (including radiology features that have been previously reported as prognostic).

Variables	Univariate		Multivariable	
	HR (95% CI)	p-value	HR (95% CI)	p-value
RPV	2.56 (1.38-4.76)	0.00288	5.11 (1.26-20.8)	0.0225
No. of locations with peritoneal disease	1.05 (1.02-1.08)	0.000649	0.989 (0.933-1.05)	0.707
Peritoneal disease in paracolic gutters	1.49 (1.19-1.86)	0.000527	1.66 (1.13-2.44)	0.00936
Peritoneal disease around liver/right upper quadrant	1.98 (1.14-3.47)	0.0163	1.80 (0.710-4.54)	0.216
Supradiaphragmatic adenopathy	3.66 (1.89-7.08)	0.000116	2.50 (1.135-5.51)	0.0230
Stage	1.63 (1.06-2.49)	0.025	1.02 (0.562-1.85)	0.952
Residual disease	1.15 (0.621-2.12)	0.658	0.700 (0.309-1.59)	0.394
Age	0.999 (0.977-1.02)	0.951	1.01 (0.982-1.05)	0.413

Figure R4. Association between RPV and number of locations with peritoneal disease, peritoneal disease in paracolic gutters, peritoneal disease around liver/right upper quadrant or supradiaphragmatic adenopathy in the HH cohort.

In addition to the previously published CT-based morphological features, we also collected a set of radiology-based features including breaching capsule, smooth outline, solid texture, homogeneous enhancement pattern, presence of thick septations, papillary projections, calcifications in the primary tumors (Fig. R5). We found RPV was higher in tumors without breaching capsule and tumors without thick septations. However, these two radiology-based features did not interact with RPV to predict overall survival in the multivariable Cox regression model (Table R5).

To conclude, our results showed the prognostic value of RPV does not derive from the radiology-based features, and RPV could potentially be used in combination with existing radiology features.

Figure R5. Association between RPV and breaching capsule, smooth outline, solid texture, homogeneous enhancement pattern, presence of thick septations, papillary projections or calcifications in the primary tumors in the HH cohort. P-values are given by two-tailed Wilcoxon rank-sum test.

Table R5. Univariate and multivariable Cox regression analysis for overall survival in the HH cohort (including semantic radiology features).

	Univariate		Multivariable		
	Variables	HR (95% CI)	p-value	HR (95% CI)	p-value
HH cohort (n=124)	RPV	2.85 (1.77-4.57)	1.48x10 ⁻⁵	3.25 (1.81-5.82)	7.47x10 ⁻⁵
	Breaching Capsule	1.08 (0.576-2.03)	0.81	0.678 (0.331-1.39)	0.288
	Thick septations	0.727 (0.403-1.31)	0.291	0.838 (0.427-1.64)	0.606
	Stage	1.63 (1.06-2.48)	0.025	1.46 (0.936-2.28)	0.0949
	Residual disease	1.15 (0.621-2.12)	0.658	1.09 (0.571-2.07)	0.798
	Age	0.999 (0.977-1.02)	0.951	0.999 (0.973-1.03)	0.936

We have included the following paragraphs in the manuscript to discuss these points:

In the Results section ‘Radiomic prognostic vector predicts survival’ (Manuscript, Page 6):

“Notably, RPV possessed a better prognostic power when compared to the existing prognostic markers including CA125 and the transcriptome-based molecular subtype and potentially synergises with existing CT-based morphological approaches (Supplementary Table 7; Supplementary Note 1).”

In the Supplementary Note 1 (Page 1):

“RPV predicts survival in synergy with existing CT-based morphological approaches

Although there is no clinically-approved approach to predict overall survival for HGSOc patients based on CT scans, a few CT characteristics summarised by radiologists were linked to prognosis in HGSOc³. To compare the prognostic performance of RPV with the existing radiology-based features,

we collected four features (Number of locations with peritoneal disease; Peritoneal disease in paracolic gutters; Peritoneal disease around liver/right upper quadrant; Supradiaphragmatic adenopathy) that were previously described to predict survival in HGSO³.

We firstly confirmed that all of these four radiology features associated with overall survival. In the multivariable Cox regression model, RPV was found associated with overall survival independent of the four radiology features (Supplementary Table 8). Interestingly, two other radiology features (Peritoneal disease in paracolic gutters and Supradiaphragmatic adenopathy) remained significant as covariates together with RPV, suggesting that a radiomics-based approach could potentially be used in combination with conventional radiology features. Furthermore, we did not observe any significant correlation between RPV with any of the prognostically-relevant radiology features, which further confirms that they are independent of each other (Supplementary Fig. 5).

In addition to the prognostically relevant radiology features, we also collected a set of radiology-based features including breaching capsule, smooth outline, solid texture, homogeneous enhancement pattern, presence of thick septations, papillary projections, and calcifications in the primary tumors (Supplementary Fig. 6). We found that RPV was higher in tumors without breaching capsule and tumors without thick septations. However, these two radiology-based features did not interact with the prognostic value of RPV in the multivariable Cox regression model (Supplementary Table 9).

CT scan is currently the standard of care imaging modality for HGSO patients and only a small proportion of patients will have PET, MRI or ultrasound, which makes CT the most attractive imaging modality for HGSO.

To conclude, our results suggest that RPV does not derive from the CT-based morphological features, but could potentially synergise with these radiology-based approaches.”

In the Discussion section (Manuscript, Page 11):

“The prognostic model we propose is simple, built solely on the information extracted from a patient’s routine preoperative CT scan at the presentation of the disease and hence readily accessible without additional costs or time delays, knowing that majority of the HGSO patients will have CT scans prior to the treatment (compared to PET, MRI or ultrasound).”

The midterm value of this signature is questionable due to the immense advances in representation learning using deep neural networks. Such approaches circumvent the manual and biased definition of image features and learn these from data while maximizing the correlation with the supervision label (here survival). This study would be significantly strengthened with inclusion of a benchmark against such procedures. Transfer learning approaches could be used to deal with the limited number of patients (see also Thrun et al. Nature, 2017).

The reviewer’s comment is well-taken. However, we believe that this may not strictly apply to the methodology we present in this study. We will highlight, and then elaborate below, what we think are predominately the main issues with a direct application of a Convolutional neural network (CNN) to our current problem:

- Images are in 3D, while transfer learning is predominately applicable to 2D images;
- small sample size.

3D images

The current applicability of transfer learning is for 2D images; this is because of the availability of large computer vision datasets (e.g. ImageNet) that contain millions of labelled 2D images. Trained CNNs with a variety of architectures (e.g. VGG, ResNet, DenseNet, GoogleNet Inception) have been trained on ImageNet and have been useful for transfer learning applications on 2D images.

Large datasets of 3D labelled medical images are difficult to obtain, and as far as we are aware, there are no publicly available 3D neural networks that have been trained on large datasets of 3D images. This makes it difficult to directly apply transfer learning to 3D images.

While some have considered extracting features from a single 2D slice (e.g. in reference⁴, the slice with largest tumor area is extracted), this does not fully solve the problem of which slice to select in tumors that have a large volume. Ovarian cancer is heterogeneous, and the appearance of texture from one slice to the next can be different. Different texture appearances lead to different values of extracted features. It is therefore best to use features extracted from the whole volume instead of a single slice. The engineered features that we have extracted are from the whole volume.

Small sample size

The reviewer queries whether we could pose the problem in a conventional machine learning way where we try to predict survival directly from the images using CNN; by doing so, image features are learnt automatically. We did not consider this approach because CNNs are well known to require a large number of training images which we did not have.

A related issue to that above is that direct application of a neural network results in a large number of features (e.g. reference⁴ uses a pre-trained network to extract 98304 features from a single slice by CNN). As the reviewer is well aware, the small sample size, coupled with many features poses a problem for data analysis, due to the phenomena known as the curse of dimensionality in machine learning.

In contrast, Thrun et al. (Nature, 2017) start off with a much larger dataset (129,450 images) of 2D clinical images in colour and train a GoogleNet Inception network with transfer learning. In our case, we have less than 300 3D images.

While we do believe that deep learning will play a big part in the future of radiomics, we think that there are still a few hurdles, as mentioned above, that prevent us from applying it to our clinical question. Manually-engineered (ME) features are currently the main component of radiomics analysis, backed by the current literature in the field. As far as we are aware, such radiomic features have not been previously applied to a multi-center ovarian tumour dataset such as ours.

For completeness, we have added the following statement to the Discussion section (Manuscript, Page 11-12).

“We employed manually-engineered features as the main component of radiomic analyses; this approach is backed by the current literature in the field⁵. While some studies⁴ have started to investigate the application of deep learning for radiomics via the extraction of thousands of deep features from convolutional networks, the small sample size coupled with the curse of dimensionality in machine learning pose a hurdle for proper evaluation of deep learning features for radiomic analysis. The availability of thousands of annotated or segmented 3D medical datasets

would allow for a more robust evaluation and open the possibility for applying transfer learning on 3D medical images, as is currently done with 2D images⁶⁻⁸.”

Reference:

1. Ataseven, B. et al. Prognostic impact of debulking surgery and residual tumor in patients with epithelial ovarian cancer FIGO stage IV. *Gynecologic oncology* **140**, 215-220 (2016).
2. Fotopoulou, C. et al. HIPEC: HOPE or HYPE in the fight against advanced ovarian cancer? *Annals of oncology : official journal of the European Society for Medical Oncology / ESMO* (2018).
3. Vargas, H.A. et al. Radiogenomics of High-Grade Serous Ovarian Cancer: Multireader Multi-Institutional Study from the Cancer Genome Atlas Ovarian Cancer Imaging Research Group. *Radiology* **285**, 482-492 (2017).
4. Lao, J. et al. A Deep Learning-Based Radiomics Model for Prediction of Survival in Glioblastoma Multiforme. *Sci Rep* **7**, 10353 (2017).
5. Aerts, H.J.W.L. et al. Decoding tumour phenotype by noninvasive imaging using a quantitative radiomics approach (vol 5, pg 4006, 2014). *Nat Commun* **5** (2014).
6. Esteva, A. et al. Dermatologist-level classification of skin cancer with deep neural networks. *Nature* **542**, 115-118 (2017).
7. De Fauw, J. et al. Clinically applicable deep learning for diagnosis and referral in retinal disease. *Nature Medicine*, 1 (2018).
8. Titano, J.J. et al. Automated deep-neural-network surveillance of cranial images for acute neurologic events. *Nature Medicine*.

Reviewers' Comments:

Reviewer #1:

Remarks to the Author:

NCOMMS-18-09461A

A Mathematical descriptor of tumor mesoscopic structure...

Many thanks for addressing the issues highlighted in the previous reviews. There are two outstanding areas of concern:

1) As requested the authors have looked at performance status in the 82 HH cohort patients for which the information is available. The univariate and multivariate analysis of standard prognostic factors are presented in Table R1 and there are further data pertaining to conventional prognostic factors (except performance status) in Table 2 of the manuscript. The problem is that table R1 clearly demonstrates the markedly abnormal demographic characteristics of the group. There are virtually no ovarian cancer studies that have not reported stage, residual disease, age and performance status as being of prognostic significance, particularly in univariate analysis (as exemplified by the TCGA data in table R3). To some extent the data in Table 2 are also of concern. Thus, RPV is derived from data sets that are not representative of ovarian cancer. The authors acknowledge and discuss this issue. However, this concern does challenge the relevance and therefore the potential of RPV.

2) The validation of eRPV is through direct comparison with RPV (fig 3d) or by molecular analogy (fig 3e). The only really direct validation of eRPV is figure 3d and this is the problem. Forty-seven patients' RPV and eRPV data are shown in figure 3d and it is clear that there is binary consistency i.e. when RPV is positive, eRPV is positive and vice versa. However, there is no apparent quantitative correlation between eRPV and RPV when the values of the two parameters are compared i.e. if RPV was plotted on the x-axis and eRPV on the y-axis. This can be seen by comparing the height of the light blue and dark blue boxes for each patient's data. As this is the only direct validation evidence of eRPV as a surrogate for RPV there remains concern about the validity and hence utility of eRPV.

Otherwise the authors have addressed my other points.

Reviewer #2:

None

1) As requested the authors have looked at performance status in the 82 HH cohort patients for which the information is available. The univariate and multivariate analysis of standard prognostic factors are presented in Table R1 and there are further data pertaining to conventional prognostic factors (except performance status) in Table 2 of the manuscript. The problem is that table R1 clearly demonstrates the markedly abnormal demographic characteristics of the group. There are virtually no ovarian cancer studies that have not reported stage, residual disease, age and performance status as being of prognostic significance, particularly in univariate analysis (as exemplified by the TCGA data in table R3). To some extent the data in Table 2 are also of concern. Thus, RPV is derived from data sets that are not representative of ovarian cancer. The authors acknowledge and discuss this issue. However, this concern does challenge the relevance and therefore the potential of RPV.

The reviewer questions the representativeness of our data set in relation to previously published ovarian cancer cohort studies, due to some unusual demographic characteristics (stage, age, residual disease and performance status) in Table R1 and Table 2. We apologise to the reviewer for not having answered his/her query to their satisfaction in the initial review round. We hope to have now made the data clearer and more transparent in the present revision.

We contend that this is a sample size issue, as only 82 cases were included in the analysis in Table R1. It is indeed a fact that for less robust variables, small patient numbers can result in inconsistent statistical outcome and interpretation, as clearly exemplified in the TCGA validation dataset of 70 patients (Table 2 of original manuscript), where none of the clinical prognostic factors (age, stage and residual disease) is significantly prognostic in uni- or multivariable analysis, as opposed to analysis of the entire TCGA data, where all of these prognostic factors are significant in the univariate analysis.

To address this point, we combined the datasets in our study and demonstrated that stage and residual disease were significant, while RPV remained the strongest prognostic factor (Table R6-R7).

Table R6. Combining HH discovery and HH validation datasets.

HH discovery + HH validation (n = 213)	Variables	Univariate		Multivariable	
		HR (95% CI)	p-value	HR (95% CI)	p-value
	RPV	2.94 (2.02-4.26)	1.54e-08	3.32 (2.16-5.10)	4.91e-08
	Stage	1.82 (1.33-2.48)	0.00017	1.75 (1.24-2.50)	0.0017
	Residual disease	1.72 (1.11-2.69)	0.0163	1.36 (0.855-2.15)	0.196
	Age ⁺	1.46 (0.951-2.24)	0.0835	1.74 (1.10-2.76)	0.0183

+ Age has been dichotomised at 60 years.

Table R7. Combining HH discovery, HH validation and TCGA validation datasets.

HH discovery+ HH validation + TCGA validation (n= 283)	Univariate			Multivariable	
	Variables	HR (95% CI)	p-value	HR (95% CI)	p-value
	RPV	3.11 (2.22-4.35)	4.34e-11	3.25 (2.19-4.81)	4.69e-09
	Stage	1.82 (1.38-2.41)	2.51e-05	1.71 (1.24-2.37)	0.00115
	Residual disease	1.67 (1.12-2.51)	0.0127	1.34 (0.879-2.04)	0.174
	Age ⁺	1.40 (0.957-2.06)	0.0827	1.54 (1.02-2.32)	0.0385

* Age has been dichotomised at 60 years.

Age: It should be noted that we had originally analysed the parameter “age” as a continuous variable; however, in the vast majority of surgical ovarian cancer studies the parameter “age” is often dichotomised to a certain cut off of, usually 60 or 65 years. When we adapt our data to a cut off of 60 years, as expected, age is significant in the multivariable analysis with a HR of 1.54-1.74 in the combined datasets. Although age is only borderline significant (p= 0.083) in the univariate analysis, we believe this is due to its relative weaker prognostic power. Indeed, a review of several studies in respectable journals shows that while some studies demonstrate age as being significant in univariate and/or multivariate analyses, several studies demonstrate age as being non-significant. Please find a summary of these studies in Table R8 below (not included in article).

Table R8. Literature review of age as prognostic factor in ovarian cancer studies.

Study	Sample size	Univariate		Multivariable	
		HR (95% CI)	p-value	HR (95% CI)	p-value
Matsuzaki et al. ¹	327	1.01 (0.98-1.03)	0.12	1.02 (0.99-1.08)	0.81
Borgonio et al. ²	132	1.02 (1.00-1.04)	0.079	1.026 (1.00-1.05)	0.053
Paju et al. ³	119	1.80 (0.99–3.24)	>0.05	Not reported	Not reported
Gallagher et al. ⁴	110	1.67 (0.81–3.44)	0.13	Not reported	Not reported
Spentzos et al. ⁵	68	Not reported	0.01	Not reported	0.27
Zhang et al. ⁶	74	Not reported	Not reported	0.83 (0.38–1.81)	ns
Merritt et al. ⁷	111	Not reported	Not reported	1.01 (0.99–1.03)	0.43

Based on these analyses we content that our datasets are representative, and we believe RPV is not only applicable to general ovarian cancer population but possesses better prognostic power than conventional prognostic factors.

To address the reviewer’s comment, we have now updated Table 2 in the revised manuscript, including the combined analysis (Table R6) together with comments on dichotomisation of age as a footnote in the table. See page 36 in the revised manuscript.

Table 2. Summary of Cox regression analysis of RPV in three datasets.

		Univariate		Multivariable	
Variables		HR (95% CI)	p-value	HR (95% CI)	p-value
HH discovery (n= 136)	RPV	4.08 (2.48-6.71)	3.37e-08	3.86 (2.30-6.46)	3.04x10 ⁻⁷
	Stage	2.03 (1.37-3.00)	0.000426	1.88 (1.24-2.86)	0.00305
	Residual disease	1.75 (1.03-2.99)	0.0393	1.40 (0.803-2.44)	0.235
	Age ⁺	1.25 (0.741-2.11)	0.404	1.47 (0.865-2.51)	0.154
HH validation (n= 77)	RPV	2.05 (1.01-4.18)	0.0485	5.08 (1.03-25.2)	0.0465
	Stage	1.32 (0.775-2.24)	0.309	1.32 (0.664-2.64)	0.425
	Residual disease	1.78 (0.777-4.08)	0.173	1.28 (0.514-3.21)	0.593
	Age ⁺	2.10 (0.940-4.68)	0.0704	3.44 (1.19-9.94)	0.0228
HH cohort combined* (n= 213)	RPV	2.94 (2.02-4.26)	1.54x10 ⁻⁸	3.32 (2.16-5.10)	4.91x10 ⁻⁸
	Stage	1.82 (1.33-2.48)	0.00017	1.75 (1.24-2.50)	0.0017
	Residual disease	1.72 (1.11-2.69)	0.0163	1.36 (0.855-2.15)	0.196
	Age ⁺	1.46 (0.951-2.24)	0.0835	1.74 (1.10-2.76)	0.0183
TCGA validation (n= 70)	RPV	4.94 (2.06-11.8)	0.00034	6.21 (2.06-18.7)	0.00117
	Stage	1.75 (0.913-3.34)	0.0921	1.03 (0.309-3.44)	0.960
	Residual disease	1.34 (0.480-3.74)	0.576	1.45 (0.414-5.05)	0.564
	Age ⁺	1.08 (0.435-2.66)	0.874	0.500 (0.154-1.63)	0.249

* Age has been dichotomised at 60 years.
* Combining HH discovery and HH validation datasets.

Performance status: The Reviewer’s comment on performance status stems from our original uni- and multivariable analyses which showed that performance status was not significantly prognostic. Unfortunately, we have the performance status of only 62 out of the total 213 HH cohort patients, and less than 20 of them had a performance status >1; the patients were operated on with cytoreductive and not palliative intent and so they all had a reasonable performance status to be able to sustain major/radical debulking surgery. For that reason, any statistical conclusions will not be valid due to the very small sample size, and for accuracy, we have therefore excluded this variable from the multivariable analysis to avoid misinterpretation in the presence of insufficient data. Performance status is another variable that is less often recorded in ovarian cancer prognostic marker studies (as opposed to therapeutic clinical trial studies). Notably all the referenced studies in Table R8 did not record performance status¹⁻⁸. To address the reviewers comment we have included the following statement:

“Age, stage and post-operative residual disease were significantly associated with OS in either uni- or multivariable analysis in the combined HH cohort while RPV remained the strongest prognostic factor, suggesting RPV is prognostic in a representative HGSOC cohort. RPV was also found associated with OS independent of performance status in a subset of patients (Supplementary Table 6). We excluded performance status from the multivariable analysis to avoid misinterpretation in the presence of insufficient data, given that we only had the performance status of 62 out of the total

213 patients in the HH cohort, and less than 20 of them had a performance status >1. For that reason, any statistical conclusions relating to performance status will not be valid due to the very small sample size.” (See page 6 in the manuscript).

2) The validation of eRPV is through direct comparison with RPV (fig 3d) or by molecular analogy (fig 3e). The only really direct validation of eRPV is figure 3d and this is the problem. Forty-seven patients’ RPV and eRPV data are shown in figure 3d and it is clear that there is binary consistency i.e. when RPV is positive, eRPV is positive and vice versa. However, there is no apparent quantitative correlation between eRPV and RPV when the values of the two parameters are compared i.e. if RPV was plotted on the x-axis and eRPV on the y-axis. This can be seen by comparing the height of the light blue and dark blue boxes for each patient’s data. As this is the only direct validation evidence of eRPV as a surrogate for RPV there remains concern about the validity and hence utility of eRPV.

We totally agree with the reviewer that this point is essential to validate the correlation between eRPV and RPV. The direct, quantitative correlation between eRPV and RPV in the TCGA dataset was plotted in Supplementary Figure 10c. We apologise for not adequately referencing this point in the manuscript. We have now done that with this revision.

Supplementary Figure 10c. Correlation between eRPV and RPV in the TCGA validation dataset. Pearson correlation coefficient and p-value is indicated.

To address the reviewer’s comment, we have now indicated the following within the body of the main manuscript:

“eRPV strongly correlated with RPV ($r = 0.720$) in the TCGA validation dataset and significantly interacted with RPV in the Cox regression model (Supplementary Figure 10c).” (See page 7 in the manuscript.)

1. Matsuzaki, H. et al. Plasma bikunin as a favorable prognostic factor in ovarian cancer. *J Clin Oncol* **23**, 1463-1472 (2005).
2. Borgono, C.A. et al. Human kallikrein 8 protein is a favorable prognostic marker in ovarian cancer. *Clinical Cancer Research* **12**, 1487-1493 (2006).
3. Paju, A. et al. Expression of trypsinogen-1, trypsinogen-2, and tumor-associated trypsin inhibitor in ovarian cancer: Prognostic study on tissue and serum. *Clinical Cancer Research* **10**, 4761-4768 (2004).
4. Gallagher, D.J. et al. Survival in epithelial ovarian cancer: a multivariate analysis incorporating BRCA mutation status and platinum sensitivity. *Ann Oncol* **22**, 1127-1132 (2011).
5. Spentzos, D. et al. Gene expression signature with independent prognostic significance in epithelial ovarian cancer. *J Clin Oncol* **22**, 4700-4710 (2004).
6. Zhang, L. et al. Intratumoral T cells, recurrence, and survival in epithelial ovarian cancer. *New Engl J Med* **348**, 203-213 (2003).
7. Merritt, W.M. et al. Dicer, Drosha, and Outcomes in Patients with Ovarian Cancer. *New Engl J Med* **359**, 2641-2650 (2008).
8. Bagnoli, M. et al. Development and validation of a microRNA-based signature (MiROvaR) to predict early relapse or progression of epithelial ovarian cancer: a cohort study. *The Lancet Oncology* **17**, 1137-1146 (2016).

Reviewers' Comments:

Reviewer #1:

Remarks to the Author:

The authors have addressed my remaining concerns